# Part-level Semantic-guided Contrastive Learning for Fine-grained Visual Classification

**Zhijian Lin**
Guangzhou Institute of Technology
Xidian University
Guangzhou, China
23021211252@stu.xidian.edu.cn

**Hong Han**
School of Electronic Engineering
Xidian University
Xi'an, China
hanh@mail.xidian.edu.cn

## Abstract

Fine-Grained Visual Classification (FGVC) aims to distinguish visually similar subcategories within a broad category, and poses significant challenges due to subtle inter-class differences, large intra-class variations, and data scarcity. Existing methods often struggle to effectively capture both part-level detail and spatial relational features, particularly across rigid and non-rigid object categories. To address these issues, we propose Part-level Semantic-guided Contrastive Learning (PSCL), a novel framework that integrates three key components. (1) The Part Localization Module (PLM) leverages clearCLIP to enable text-controllable region selection, achieving decoupled and semantically guided spatial feature extraction. (2) The Multi-scale Multi-part Branch Progressive Reasoning (MMBPR) module captures discriminative features across multiple parts and scales, while reducing inter-branch redundancy. (3) The Visual-Language Contrastive Learning based on Multi-grained Text Features (VLCL-MG) module introduces intermediate-granularity category concepts to improve feature alignment and inter-class separability. Extensive experiments on five publicly available FGVC datasets demonstrate the superior performance and generalization ability of PSCL, validating the effectiveness of its modular design and the synergy between vision and language. Code is available at: https://github.com/joker-lin9/PSCL.

## 1 Introduction

Fine-Grained Visual Classification (FGVC) aims to accurately distinguish between subcategories that belong to the same high-level category yet exhibit subtle visual differences. Typical applications include the classification of bird species (Wah et al., 2011; Van Horn et al., 2015), car brands (Krause et al., 2013), and aircraft (Maji et al., 2013) models. As FGVC focuses on fine-level distinctions within specific domains, it has demonstrated unique practical value—distinct from general visual classification tasks—in areas such as intelligent transportation, medical image analysis, and ecological environment monitoring. However, FGVC remains a challenging task due to factors such as low inter-class variance, high intra-class variance, a large number of categories, and data scarcity.

We observe that existing models exhibit notable feature preferences when processing rigid and non-rigid objects. We argue that FGVC tasks require the modeling of two key types of features: (1) part-level fine-grained features that capture detailed local differences and (2) spatial relational features that describe inter-class differences in spatial structure. For rigid objects, inter-class differentiation is often affected by external factors such as viewpoint variation and occlusion. In contrast, non-rigid objects tend to exhibit more significant posture variations, leading to greater uncertainty in their spatial structural features. Different model architectures vary considerably in their capacity to capture these two types of features.

Some existing works have consciously incorporated mechanisms for modeling spatial structural features. For example, CAP (Behera et al., 2021) captures spatial relations through region consistency integration, while SFETrans (Yu et al., 2025) extracts spatial features via phase spectrum analysis.

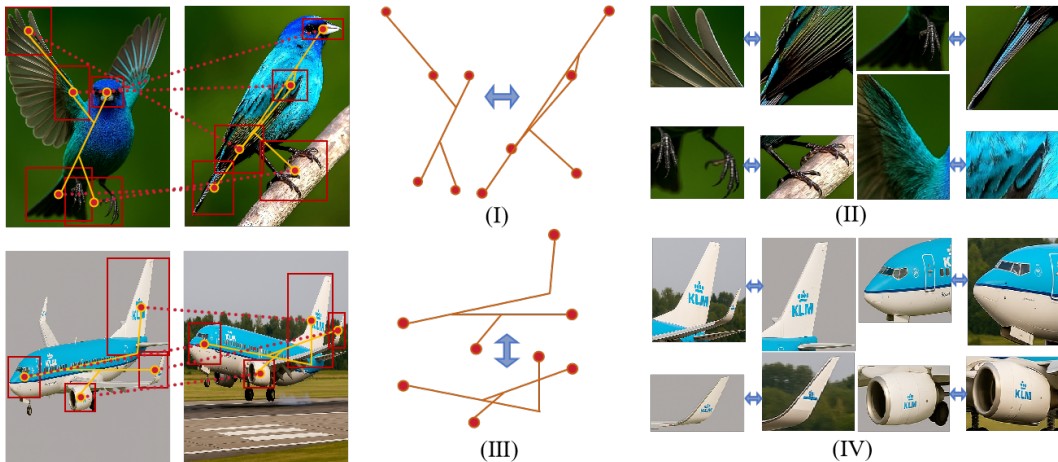

Figure 1: Two types of critical features in rigid and non-rigid objects. (I) Spatial deformation in non-rigid objects (e.g., birds) due to articulated motion; (II) Diverse part-level details in non-rigid objects; (III) Stable spatial structure in rigid objects (e.g., airplanes); (IV) Consistent part-level appearance in rigid objects.

These methods have demonstrated effectiveness in improving classification performance for rigid objects. However, the core objective of FGVC lies in accurately modeling subtle inter-class differences. Since spatial relational features often rely on matching shared regions across categories, they may conflict with the precise representation of part-level details—particularly for non-rigid objects—potentially weakening the model's ability to focus on critical parts. Furthermore, current models generally adopt a unified strategy for designing part-based branches across all categories, overlooking the homogeneity of part-level details among similar categories. This can lead to misclassifications and redundant representations across branches.

To address these issues, we propose a novel framework called Part-level Semantic-guided Contrastive Learning (PSCL). This model introduces a Part Localization Module (PLM), which leverages clearCLIP (Lan et al., 2024) as an auxiliary component to enable text-guided region selection, thereby achieving effective decoupling of feature region selection and feature representation. Additionally, we design a Multi-scale Multi-part Branch Progressive Reasoning (MMBPR) module, where part-based branches represent fine-grained features of individual parts, while a global branch adaptively integrates features based on spatial relations. Through progressive reasoning, MMBPR enables each branch to refine its feature representations across multiple scales.

During the multi-scale feature fusion stage, we further design the Reverse-key Scale-aware Attention Fusion Module (ReSAF) to suppress the influence of high-level features on semantically similar regions at lower levels, thereby encouraging the model to extract information from less similar areas. This effectively mitigates feature redundancy among branches.

Finally, in the classification phase, we introduce a novel Visual-Language Contrastive Learning based on Multi-grained Text Features (VLCL-MG) module. By incorporating intermediate-level category concepts, this module leverages prior knowledge to aggregate fine-grained categories into semantically coherent mid-level groups, promoting more meaningful clustering of similar subcategories in the feature space.

Our main contributions can be summarized as follows:

- We propose a Part Localization Module (PLM) that enables text-controllable spatial feature extraction via clearCLIP;
- We design a Multi-scale Multi-part Branch Progressive Reasoning (MMBPR) module to reduce feature redundancy and enhance part-level and global representations;
- We introduce a Visual-Language Contrastive Learning module based on Multi-grained Text Features (VLCL-MG) to improve the semantic alignment of visually similar subcategories;
- Extensive experiments on five publicly available FGVC datasets validate the effectiveness and generalization ability of our proposed PSCL framework.

## 2 RELATED WORK

### 2.1 FINE-GRAINED VISUAL CLASSIFICATION

Fine-grained visual classification (FGVC) methods primarily focus on capturing subtle inter-class differences through refined feature representation and part localization. Early feature representation approaches relied on high-level features (Lin et al., 2015; Zheng et al., 2019; Sun et al., 2020), later incorporating multi-scale fusion techniques such as AP-CNN (Ding et al., 2021) and PMG (Du et al., 2020), as well as attention-based mechanisms like MA-CNN (Zheng et al., 2017), OSME (Sun et al., 2018), and Transformer-based methods such as TransFG (He et al., 2022) and CAMF (Miao et al., 2021). MDCM (Zhang et al., 2025) introduces a multi-scale ViT framework that improves fine-grained bird recognition by activating, selecting, and aggregating discriminative cues across scales. Part localization methods identify discriminative regions through cropping and scaling strategies. This line of work aims to locate category-relevant regions within the input image by analyzing attention maps generated by the backbone network. The identified regions are then cropped and reprocessed to retain high-resolution, fine-grained details that are critical for classification. This strategy explicitly extracts spatial structural features by emphasizing salient parts, often leading to superior classification performance. While early approaches like Part-based R-CNN (Zhang et al., 2014) and Pose Normalized CNN (Branson et al., 2014) relied on strong supervision, recent methods have shifted to weak supervision for better scalability. Notable examples include MGE-CNN (Zhang et al., 2019), P2P-Net (Yang et al., 2022), CAP (Behera et al., 2021), TBMSL-Net (Zhang et al., 2021), and PART (Zhao et al., 2021), which explore part-level semantics via multi-scale learning, context modeling, or Transformer-based architectures. CSQA-Net (Xu et al., 2025) introduces a Part Navigator module to assign saliency scores to different image regions, enabling discriminative region segmentation without strong part annotations.

### 2.2 VISION-LANGUAGE LEARNING

Vision-language models (VLMs), particularly CLIP (Radford et al., 2021), have demonstrated strong potential in open-vocabulary tasks by learning joint representations from large-scale image-text pairs. While early FGVC-related works using CLIP (Li et al., 2023; Wang et al., 2023b) emphasized alignment between descriptive text and novel categories, MP-FGVC (Jiang et al., 2024) introduced CLIP to closed-set FGVC by leveraging multimodal prompts to enhance category discrimination. For region-level tasks, CLIP's utility has been extended to open-vocabulary segmentation. MaskCLIP (Zhou et al., 2022) revealed that dense patch-level features from CLIP's attention layers could be aligned with textual representations. Building on this, ClearCLIP (Lan et al., 2024) demonstrates that by removing residual connections in CLIP, enabling self-attention, and eliminating the feed-forward network, open-vocabulary semantic segmentation can be achieved directly without additional training. We empirically demonstrate that ClearCLIP is also effective for part-level semantic concepts.

## 3 APPROACH

The proposed PSCL architecture is illustrated in fig. 2. In the visual pathway, the input image is first processed separately by the backbone and ClearCLIP. ClearCLIP generates part masks by computing matching scores and applying channel selection, while the backbone produces multi-scale features. For single-scale backbones such as ViT, these features can be regarded as multi-level representations extracted from different transformer layers, which serves an equivalent role in our framework and does not affect the overall conclusion. The two outputs are combined using the Hadamard product to obtain multi-scale part-level features, forming the Part Localization Module. The designed Multi-scale Multi-part Branch Progressive Reasoning module processes the resulting visual features, progressively enhancing the model's confidence in its predictions from low-level to high-level features. This confidence enhancement is achieved through a combination of hyperparameters for contrastive loss weights across different scales and noise parameters. In the text pathway, contrastive loss leverages intermediate-grained textual priors as input, generating multi-grained textual features for different categories. These features are then rearranged and restructured to produce multi-grained textual representations for each fine-grained label.

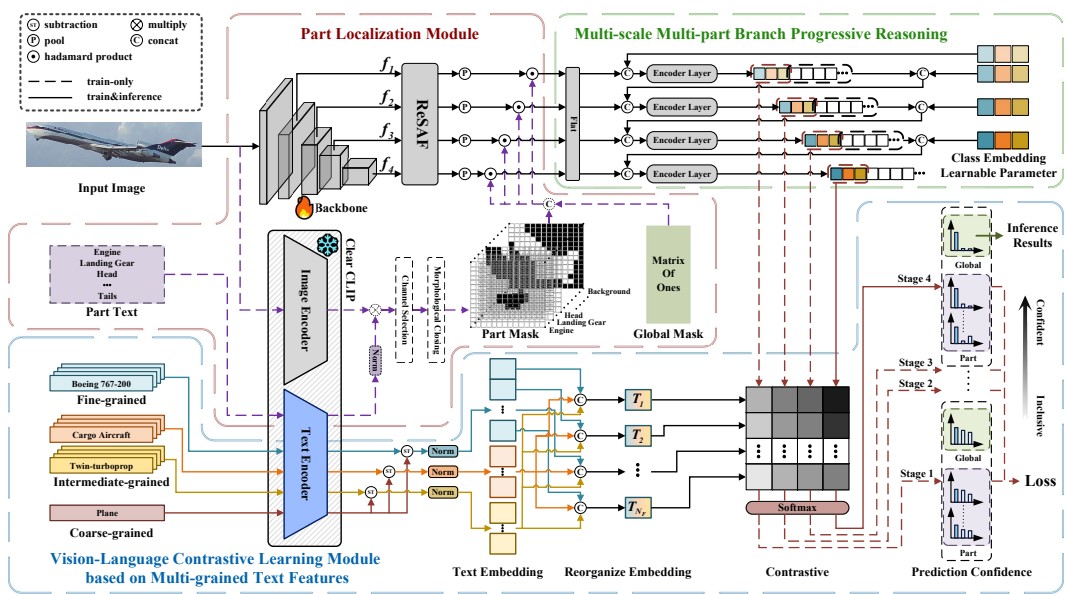

Figure 2: Detailed illustration of Part-level Semantic-guided Contrastive Learning model (PSCL).

## 3.1 PART LOCALIZATION MODULE

The proposed Part Localization Module (PLM) is designed to address the conflicting requirements of modeling fine-grained part-level features and spatial relational features in FGVC. This conflict is particularly pronounced for non-rigid objects, where posture variation undermines the stability of spatial structures and affects precise part representation. To resolve this, PLM processes the input image $\mathbf{x} \in \mathbb{R}^{C \times H \times W}$ through two separate branches: one for capturing difference-aware features and the other for localizing discriminative parts, enabling more effective and targeted feature learning across object types.

The branch responsible for representing differences processes the input $\mathbf{x}$ to produce multiscale features, with features denoted as $f_s \in \mathbb{R}^{C_s \times H_s \times W_s}$ across multiple stages. When low-level features are less relevant for classification, only higher stages may be selected, such that

$$s \in \{s_{\min}, \ldots, 4\}, \quad s_{\min} \geq 1, \tag{1}$$

where $s_{\min}$ denotes the earliest stage used, which can be adjusted based on task requirements.

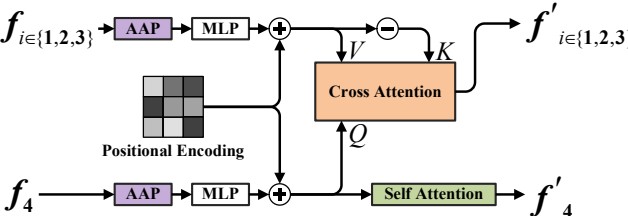

Figure 3: Illustration of ReSAF.

The resulting $f_s$ is then passed into the Reverse-key Scale-aware Attention Fusion Module (ReSAF) to suppress redundant channel representations across scales, as illustrated in fig. 3. In the figure, AAP denotes Adaptive Average Pooling, and the positional encoding is implemented as a learnable parameter. By flipping key vector directions, ReSAF inverts similarity scores, guiding high-level queries to attend away from similar low-level regions. This contrastive attention promotes the extraction of complementary.

The branch responsible for identifying the parts of interest is built upon the ClearCLIP backbone. The input image $\mathbf{x}$ is encoded by the image encoder $f_{\text{img}}$, producing patch-level image features $\mathbf{F}_{\text{img}} = f_{\text{img}}(\mathbf{x}) \in \mathbb{R}^{H \times W \times d}$. Similarly, the textual prompts corresponding to $N$ parts, denoted as $\mathbf{T} = \{T_1, T_2, \ldots, T_N\}$, are processed by the text encoder $f_{\text{text}}$, yielding part-specific text feature representations $\mathbf{F}_{\text{text}} = \{f_{\text{text}}(T_1), f_{\text{text}}(T_2), \ldots, f_{\text{text}}(T_N)\} \in \mathbb{R}^{N \times d}$. To align image patches with text descriptions, the similarity tensor $\mathbf{S}$ is computed via matrix multiplication.

$$S = F_{\text{img}} F_{\text{text}}^{\top}, S \in R^{H \times W \times N}. \tag{2}$$

To generate the final part mask $\mathbf{M}$, the indices of the maximum similarity scores across the $N$ channels are first determined as

$$\text{max\_indices} = \arg \max_{j \in \{1, \ldots, N\}} \mathbf{S}[j]. \tag{3}$$

Using these indices, a one-hot-like tensor is constructed:

$$\mathbf{S}[j] = \begin{cases} 1, & j = \text{max\_indices}, \\ 0, & \text{otherwise}. \end{cases} \tag{4}$$

The one-hot-like tensor undergoes morphological refinement by first applying dilation to expand regions, followed by erosion to refine connectivity and remove noise:

$$\mathbf{M} = (S \oplus \mathcal{K}) \ominus \mathcal{K}, \tag{5}$$

where $\mathcal{K}$ denotes the structuring element (kernel), instantiated as a $3 \times 3$ kernel in our implementation; $\oplus$ represents the morphological dilation operator; and $\ominus$ denotes the erosion operator.

The multi-scale multi-part features $G_{s,n'}$ can be expressed as:

$$G_{s,n'} = \text{concat}\left(f'_s \odot \mathbf{M}_{s,n}, f'_s \odot \mathbf{1}\right), n \in \{1, 2, \ldots, N\}, n' \in \{1, 2, \ldots, N+1\}, \tag{6}$$

where $\odot$ denotes the Hadamard product, $\text{concat}(\cdot)$ denotes the concatenation operation, $\mathbf{1}$ is a matrix of ones, representing the global mask, $f'_s \odot \mathbf{1}$, captures the global features. The global features are subsequently processed by the global branch, which adaptively aggregates part-level information according to spatial relationships, aiming to model spatial relational features.

## 3.2 BRANCH PROGRESSIVE REASONING

Our proposed Multi-scale Multi-part Branch Progressive Reasoning (MMBPR) module extends the multi-scale reasoning framework introduced by PMG (Du et al., 2020) and PART (Zhao et al., 2021). It progressively enhances supervision from low-level to high-level feature branches. In addition to multi-scale progressive reasoning, the framework integrates part-level branches for fine-grained local modeling and a global branch for capturing spatial relational representations via part aggregation.

Although multi-part-branch designs have become standard in FGVC, most existing methods rely on a shared feature extractor for both localization and classification, resulting in weakly grounded parts biased toward classification objectives. For example, PART (Zhao et al., 2021) selects regions via class activation maps and saliency ranking, which does not ensure consistent semantic correspondence across samples. Under occlusion or pose variation, a branch may attend to different components across images, leading to redundancy and unstable learning.

In contrast, PLM leverages text prompts and ClearCLIP-based localization to achieve semantically grounded and controllable part selection, where each branch corresponds to a predefined part. By decoupling region selection from representation learning, and deactivating occluded parts via the mask formulation in eq. (6), PLM maintains stable optimization.

Following the ViT architecture, we adopt Class Embedding Learnable Parameters to extract category-specific visual representations. We utilize three class tokens to comprehensively represent intermediate categories.

The overall input to MMBPR is $G_{s,n'}$ obtained from eq. (6), and the progressive reasoning process begins from the lowest-level features $G_{s_{\min},n'}$, as formulated below:

$$G_{s_{\min},n'} \xrightarrow{\text{flatten}} \{V_{s_{\min},n',m} \mid m \in \{1, 2, \ldots, P^2\}\}. \tag{7}$$

The flattened tokens are concatenated with class tokens:

$$Z_{s_{\min},n'} = \text{concat}([C_{\text{cls},s_{\min},n',1}; C_{\text{cls},s_{\min},n',2}; C_{\text{cls},s_{\min},n',3}], [V_{s_{\min},n',1}; \ldots; V_{s_{\min},n',P^2}]), \tag{8}$$

where $C_{\text{cls},s_{\min},n',j}$ ($j \in \{1, 2, 3\}$) are class tokens.

The sequence $Z$ is then processed through an Encoder Layer:

$$\begin{aligned} Z'_{s_{\min},n'} &= \text{LN}(Z_{s_{\min},n'} + \text{MHSA}(Z_{s_{\min},n'})), \\ Z''_{s_{\min},n'} &= \text{LN}(Z'_{s_{\min},n'} + \text{MLP}(Z'_{s_{\min},n'})), \end{aligned} \tag{9}$$

where $\mathrm{LN}(\cdot)$ is Layer Normalization, $\mathrm{MHSA}(\cdot)$ is Multi-Head Self-Attention, $\mathrm{MLP}(\cdot)$ is a feed-forward neural network, and the model weights are not shared across layers. The resulting output is:

$$Z''_{s_{\min},n'} = [C''_{\mathrm{cls},s_{\min},n',1}; C''_{\mathrm{cls},s_{\min},n',2}; C''_{\mathrm{cls},s_{\min},n',3}; V''_{s_{\min},n',1}; \ldots; V''_{s_{\min},n',P2}]. \tag{10}$$

The output $Z''_{s_{\min},n'}$ is then divided into two parts: Class tokens, representing the visual features of the categories at the stage $\mathbf{I}_{s_{\min},n'} = [C''_{\mathrm{cls},s_{\min},n',1}; C''_{\mathrm{cls},s_{\min},n',2}; C''_{\mathrm{cls},s_{\min},n',3}]$ are passed to VLCL-MG for contrastive learning. Feature tokens $[V''_{s_{\min},n',1}; \ldots; V''_{s_{\min},n',P2}]$ are forwarded to the next stage, where they are concatenated with the flattened high-level features.

By ensuring non-interference between the lower and higher branches, this design enables the higher-level feature branch to acquire stronger discriminative capabilities, thereby leading to more confident category-specific visual representations.

The process is recursively applied to the next stage:

$$Z_{s_{\min+1},n'} = \mathrm{concat}([C_{\mathrm{cls},s_{\min+1},n',1}; C_{\mathrm{cls},s_{\min+1},n',2}; C_{\mathrm{cls},s_{\min+1},n',3}],$$
$$[V_{s_{\min+1},n',1}; \ldots; V_{s_{\min+1},n',P2}], [V''_{s_{\min},n',1}; \ldots; V''_{s_{\min},n',P2}]). \tag{11}$$

This procedure is iteratively applied until the highest-level feature branch completes its reasoning process, and the reasoning is conducted in both the global branch and the part-level branches.

## 3.3 VISION-LANGUAGE CONTRASTIVE LEARNING

Our proposed Vision-Language Contrastive Learning Module based on Multi-grained Text Features (VLCL-MG) constrains the inter-class differences of visual features to align with real-world distinctions by introducing intermediate category constraints, which are primarily implemented through the model structure.

Additionally, in terms of loss computation, since FGVC tasks involve highly similar subcategories, we argue that absolute model outputs can unnecessarily enlarge inter-class distances. If the top-scoring category is incorrect, the actual category score may rank lower due to small score differences among other categories. To mitigate this, we employ label smoothing (Szegedy et al., 2016) for regularization. Meanwhile, given the limited number of samples, models are prone to overfitting, making it crucial to learn more from hard-to-classify samples. Therefore, focal loss (Lin et al., 2017) is also essential. Based on these considerations, we propose the Focal-Smooth Contrastive Loss as a complement to our model structure.

Specifically, we first obtain intermediate categories for fine-grained labels. For example, between the coarse-grained category `airplane` and the fine-grained category `Boeing 737-200`, intermediate categories include `narrow-body airliner` and `twinjet`. This expert knowledge can be efficiently obtained with minimal effort, requiring only a one-time retrieval per category rather than per-image annotation. For most datasets, we employ ChatGPT-4o for semi-automated knowledge retrieval, whereas for the NABirds dataset, we directly utilize the dataset's built-in class hierarchy.

The multi-grained textual labels corresponding to each fine-grained category are represented as

$$\mathbf{t}_{\mathrm{cls}} = \{a_{n_A}, b_{n_B}, f_{n_F}\} \in \mathbb{R}^C, \tag{12}$$

where $a_{n_A}$ and $b_{n_B}$ represent two types of intermediate categories, and $f_{n_F}$ corresponds to fine-grained categories, $C$ represents the number of fine-grained categories. These labels are processed by the ClearCLIP text encoder, yielding multi-grained text features:

$$\mathbf{t}_{n_F} = f_{\mathrm{text}}(\mathbf{t}_{\mathrm{cls}}). \tag{13}$$

To prevent text features of all grained levels from clustering too closely in the embedding space, we subtract the coarse-grained category feature $f_{\mathrm{text}}(c)$ from the multi-grained text features $\mathbf{t}_{n_F}$ and then apply normalization.

$$\mathbf{T}_{n_F} = \mathrm{norm}(\mathbf{t}_{n_F} - f_{\mathrm{text}}(coarse)). \tag{14}$$

This operation ensures a more discriminative distribution of text embeddings across different grained levels. To avoid redundant computations, all category label texts are first processed by the text

encoder, and then rearranged according to the intermediate-grained categories corresponding to each fine-grained label, as detailed in fig. 2.

The predicted probability distribution $\mathbf{P}_s$ is obtained by applying a softmax normalization over the similarity scores between the linearly projected feature $\mathbf{I}_{s,n'}$, extracted from eq. (10) at stage $s$, and the final-stage prototype representation $\mathbf{T}_{n_F}$. To ensure dimensional compatibility, each feature $\mathbf{I}_{s,n'} \in \mathbb{R}^{D_I}$ is first transformed by a learnable linear projection $\mathbf{W} \in \mathbb{R}^{D_T \times D_I}$. The class-wise probability for the $n'$-th branch is defined as:

$$\mathbf{P}_{s,n',c} = \sigma\big(\boldsymbol{\tau} \odot \big((\mathbf{W}\mathbf{I}_{s,n'})\,\mathbf{T}_{n_F}^\top\big) + \boldsymbol{\beta}\big)_c, \tag{15}$$

where $\boldsymbol{\tau} \in \mathbb{R}^C$ denotes a learnable temperature scaling vector, $\boldsymbol{\beta} \in \mathbb{R}^C$ is a learnable bias term, and $\sigma(\cdot)$ represents the softmax function applied over all classes $c = 1, \ldots, C$.

The Focal-Smooth Contrastive Loss at stage $s$, denoted as $\mathcal{FSL}_s(\mathbf{P}_s, y)$, is formulated as:

$$\mathcal{FSL}_s(\mathbf{P}_s, y) = -\sum_{n'=1}^{N+1} \sum_{c=1}^{C} (1 - P_{s,n',c})^\gamma\, \tilde{y}_{s,n',c} \log P_{s,n',c}, \tag{16}$$

where $\gamma$ is the focusing factor that adjusts the impact of misclassified examples. The smoothed target distribution $\tilde{y}_{s,n',c}$ is given by:

$$\tilde{y}_{s,n',c} = \begin{cases} 1 - \epsilon_s, & \text{if } c = y_{s,n'}, \\ \epsilon_s/(C-1), & \text{otherwise}, \end{cases} \tag{17}$$

where $\epsilon_s$ gradually decreases as the stage index $s$ increases, enabling the MMBPR module to generate progressively more confident predictions.

The final loss is defined as the weighted sum of the stage-wise losses $\mathcal{FSL}_s(\mathbf{P}_s, y)$, where the weight $\tilde{\epsilon}_s$ increases with the stage index $s$ (serving a role analogous to $\epsilon_s$, but exhibiting an opposite monotonic trend). Formally,

$$\mathcal{L}_{\text{final}} = \sum_{s=s_{\min}}^{4} \tilde{\epsilon}_s \cdot \mathcal{FSL}_s(\mathbf{P_s}, y). \tag{18}$$

At inference time, the prediction from the final stage is utilized, and the inference strategy relies solely on the global branch:

$$\mathbf{P}^{\text{inference}} = \mathbf{P}_{s=4,\,n'=0}, \tag{19}$$

in which ClearCLIP and redundant part-level branches are removed during inference, thereby enabling substantially faster computation.

## 4 EXPERIMENTS

### 4.1 EXPERIMENTAL SETUP

Table 1: Statistics of benchmark datasets.

| Dataset | Class | Train | Test |
|---|---|---|---|
| FGVC Aircraft (AIR) | 100 | 6,667 | 3,333 |
| Stanford Dogs (DOG) | 120 | 12,000 | 8,580 |
| Stanford Cars (CAR) | 196 | 8,144 | 8,041 |
| CUB-200-2011 (CUB) | 200 | 5,994 | 5,794 |
| NABirds (NAB) | 555 | 23,929 | 24,633 |

**Datasets** We comprehensively evaluate PSCL on the FGVC Aircraft (Maji et al., 2013), Stanford Dogs(Khosla et al., 2011), Stanford Cars (Krause et al., 2013), CUB-200-2011 (Wah et al., 2011) and NABirds (Van Horn et al., 2015) datasets, which are widely used FGVC benchmarks. In all experiments, we do not utilize part annotations and follow the same train/test split. The details of the five datasets are presented in table 1.

**Implementation Details** We adopt ResNet-50 (He et al., 2016), Vision Transformer (Dosovitskiy et al., 2021), and Swin Transformer (Liu et al., 2021) as the backbone architectures. The input resolutions are $448 \times 448$ for RN50 and $384 \times 384$ for Swin-B. For ViT-B, we use two input resolutions: $448 \times 448$ for fair comparison with other methods, and $518 \times 518$ as the default setting that achieves the best performance. Unless otherwise specified, ViT-B denotes the $518 \times 518$ model. The impact

of different resolutions is shown in appendix H. During training, we apply standard data augmentation techniques, including random cropping, random erasing, horizontal flipping, Gaussian blur, color jittering, and rotation. All models are trained for 100 epochs using the AdamW optimizer with a batch size of 16 and a weight decay of 0.01. The initial learning rate is set to $1 \times 10^{-4}$ for RN50 and $1 \times 10^{-5}$ for both ViT-B and Swin-B. A warm-up phase of 10 epochs is applied, and the learning rate follows a cosine annealing schedule. The focusing parameter $\gamma$ is set to 4, and the smoothing noise factor $\epsilon_s$ follows [0.4, 0.3, 0.2, 0.1], while $\tilde{\epsilon}_s$ is set to [0.1, 0.2, 0.4, 1.0]. The hyperparameter settings are summarized in table 8.

## 4.2 COMPARISON WITH OTHER METHODS

Table 2: Performance comparison on FGVC benchmark datasets (Accuracy %). The best results for each dataset are highlighted in bold.

| Method | Backbone | Input Resolution | AIR | CAR | CUB | NAB | DOG |
|--------|----------|------------------|-----|-----|-----|-----|-----|
| CMN (Deng et al., 2022) | RN50 | 448×448 | 93.8 | 94.9 | 88.2 | 87.8 | – |
| P2P-Net (Yang et al., 2022) | RN50 | 448×448 | 94.2 | 95.4 | 90.2 | – | – |
| GDSMP-Net (Ke et al., 2023) | RN50 | 448×448 | 94.4 | 95.3 | 89.9 | **89.0** | – |
| SIA-Net (Wang et al., 2023c) | RN50 | 448×448 | 94.3 | 95.5 | **90.7** | – | – |
| **PSCL (ours)** | RN50 | 448×448 | **95.1** | **95.6** | 89.1 | **89.0** | 90.1 |
| TransFG (He et al., 2022) | ViT-B | 448×448 | – | 94.8 | 91.7 | 90.8 | 92.3 |
| MpT-Trans (Wang et al., 2023a) | ViT-B | 448×448 | 92.2 | 93.8 | 92.0 | 91.3 | – |
| ACC-ViT (Zhang et al., 2024) | ViT-B | 448×448 | – | 94.9 | 91.8 | 91.4 | **92.9** |
| MP-FGVC (Jiang et al., 2024) | ViT-B | 448×448 | – | – | 91.8 | 91.0 | 91.0 |
| **PSCL (ours)** | ViT-B | 448×448 | 94.3 | 95.1 | 92.2 | 92.5 | 91.0 |
| **PSCL (ours)** | ViT-B | 518×518 | **96.5** | **96.4** | **92.3** | **93.7** | 92.3 |
| ViT-NeT (Kim et al., 2022) | Swin-B | 224×224 | – | 95.0 | 91.6 | 90.9 | 90.3 |
| TransIFC+ (Liu et al., 2023) | Swin-B | 448×448 | – | – | 91.0 | 90.9 | – |
| HERBS (Chou et al., 2023) | Swin-B | 384×384 | – | – | 92.3 | 93.0 | – |
| CSQA-Net (Xu et al., 2025) | Swin-B | 448×448 | 94.7 | **95.6** | 92.6 | 92.3 | – |
| **PSCL (ours)** | Swin-B | 384×384 | **95.3** | 95.5 | **93.0** | **93.8** | **94.7** |

We evaluate our method on five benchmark datasets using three backbone architectures and compare it with state-of-the-art models, as summarized in table 2. The results demonstrate the superior performance and strong generalization ability of PSCL across diverse FGVC benchmarks. PSCL consistently achieves state-of-the-art or highly competitive accuracy across all datasets and backbones (RN50, ViT-B, and Swin-B). It delivers substantial improvements under Transformer-based backbones, and remains competitive under the CNN-based RN50, particularly on AIR and CAR datasets. These results highlight PSCL's adaptability to different architectures and its effectiveness in capturing both local and structural discriminative cues. Furthermore, its consistent performance across datasets underscores its robustness. Notably, on the large-scale NAB dataset, the availability of an accurate and professionally curated category hierarchy enables precise intermediate-level grouping, further enhancing accuracy and demonstrating the effectiveness of the VLCL-MG module. The strong performance on multiple non-rigid datasets such as CUB and NAB demonstrates that PSCL effectively models the characteristics of non-rigid objects.

## 4.3 EFFECTIVENESS OF MODULE OPERATION

**Locating Relevant Parts** We resize eq. (2) to match the original image dimensions for visualizing the part localization results. As observed in fig. 4, the PLM structure effectively identifies the locations of the parts.

**Reverse-key Scale-aware Attention Fusion Module** To assess the effectiveness of our proposed ReSAF module, we conduct a comparative study with two alternative intermediate mechanisms: a multilayer perceptron (MLP) and cross-attention. All experiments are performed using the RN50 backbone on the AIR dataset. The quantitative results, summarized in table 3, demonstrate that ReSAF consistently outperforms the other two variants, highlighting its superior capability in capturing scale-aware feature interactions. More comparative results and analyses are provided in appendix I.

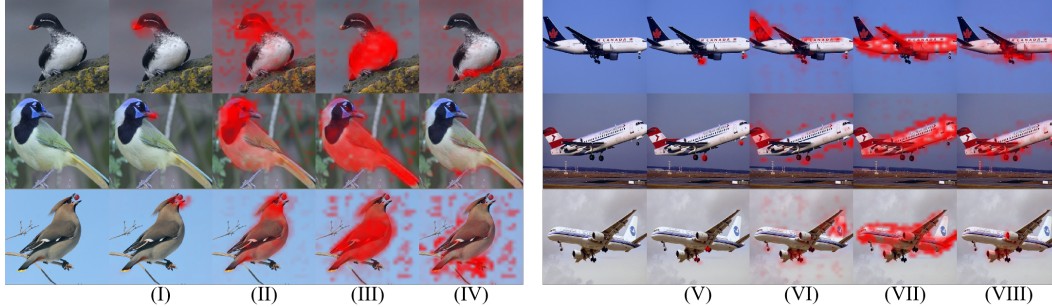

Figure 4: Part score visualization. PLM uses the following textual prompts: (I) mouth; (II) head; (III) body; (IV) foot; (V) landing gear; (VI) tail; (VII) fuselage; (VIII) engine.

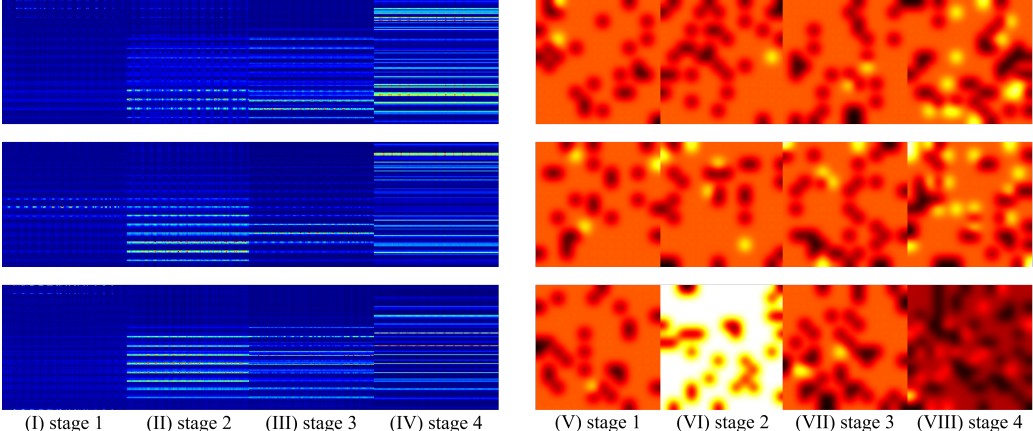

(I) stage 1     (II) stage 2     (III) stage 3     (IV) stage 4      (V) stage 1     (VI) stage 2     (VII) stage 3     (VIII) stage 4

Figure 5: Attention maps of ReSAF. (I)-(IV) show the relative attention among patches, while (V)-(VIII) present the accumulated spatial attention. It can be observed that shallow features primarily serve a complementary role for deep features.

Table 3: Performance comparison of different intermediate mechanisms on the AIR dataset. Accuracy (%) is reported.

| Intermediate Mechanism | Accuracy (%) |
|---|---|
| MLP | 94.71 |
| Cross-Attention | 94.99 |
| ReSAF (Ours) | **95.14** |

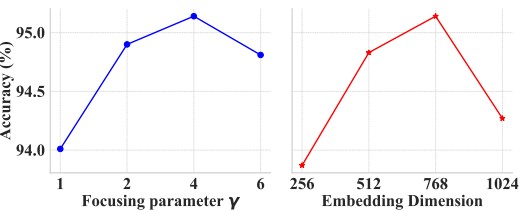

Figure 6: Accuracy (%) on the AIR dataset (RN50) under different settings. Left: focusing parameter $\gamma$; Right: encoder hidden dimension.

**Hyperparameter Selection** All hyperparameters except the learning rate were searched and selected exclusively on the AIR dataset with RN50. Results are shown in table 4, table 5 and fig. 6. The trends of $\tilde{\epsilon}_s$ and $\gamma$ suggest that progressive inference improves performance, and increasing the focus on hard samples via $\gamma$ further enhances results. We note that if hyperparameter tuning were performed specifically for a target dataset, our proposed PSCL could potentially achieve even better performance.

**Ablation Studies** The ablation results in table 6 across RN50, Swin-B, and ViT-B on CUB, AIR, and CAR verify the effectiveness of each proposed module. PLM and VLCL-MG individually yield notable gains, reflecting their strengths in part localization and semantic alignment. Because both are designed to address the same underlying issue, combining them may show diminishing marginal gains—a property rooted in the data itself. Fine-grained classes sharing the same intermediate category often exhibit similar part-level structures (e.g., Anseriformes birds with webbed feet and long necks; off-road vehicles with high chassis and traction-oriented tires), which reduces the additional benefits of stacking PLM and VLCL-MG. MMBPR further improves performance through multi-scale reasoning. Although the modules differ in their roles, all aim to enhance semantic consistency, and despite the potential overlap of their effects in low-redundancy settings, the full model consistently achieves the best results, confirming their overall complementarity.

Table 4: Accuracy (%) on AIR dataset (RN50) under smoothing noise factor $\epsilon_s$.

| $\epsilon_s$ | Accuracy (%) |
|---|---|
| $[0.0, 0.0, 0.0, 0.0]$ | 89.92 |
| $[0.6, 0.4, 0.2, 0.0]$ | 94.22 |
| $[0.7, 0.5, 0.3, 0.1]$ | 94.83 |
| $[0.4, 0.3, 0.2, 0.1]$ | **95.14** |

Table 5: Accuracy (%) on AIR dataset (RN50) under multi-scale loss weight coefficient $\tilde{\epsilon}_s$.

| $\tilde{\epsilon}_s$ | Accuracy (%) |
|---|---|
| $[0.0, 0.0, 0.0, 1.0]$ | 94.65 |
| $[0.1, 0.2, 0.2, 1.0]$ | 94.77 |
| $[0.1, 0.2, 0.4, 1.0]$ | **95.14** |
| $[0.2, 0.4, 0.4, 1.0]$ | 94.89 |

Table 6: Ablation study on three FGVC datasets using different backbones. Accuracy (%) is reported for each configuration. The best results for each column are highlighted in bold. Features are indicated by a check mark (✔) or a cross (✘).

| PLM | MMBPR | VLCL-MG | RN50 | | | Swin-B | | | ViT-B | | |
|---|---|---|---|---|---|---|---|---|---|---|---|
| | | | CUB | AIR | CAR | CUB | AIR | CAR | CUB | AIR | CAR |
| ✘ | ✘ | ✘ | 85.09 | 91.56 | 91.90 | 92.32 | 94.14 | 94.86 | 90.23 | 94.57 | 95.60 |
| ✔ | ✘ | ✘ | 88.82 | 94.54 | 95.46 | 92.68 | 94.60 | 94.74 | 91.99 | 96.13 | 96.05 |
| ✔ | ✔ | ✘ | 89.09 | 94.54 | 95.54 | 92.51 | 95.05 | 95.04 | **92.34** | 96.19 | 96.17 |
| ✘ | ✘ | ✔ | 87.90 | 94.39 | 95.32 | 92.65 | 94.87 | 95.51 | 90.94 | 95.08 | 96.26 |
| ✔ | ✔ | ✔ | **89.13** | **95.14** | **95.59** | **93.01** | **95.32** | **95.54** | **92.34** | **96.48** | **96.44** |

## 4.4 TRAINING AND INFERENCE EFFICIENCY

**Additional Computational Cost During Training.** Although PSCL introduces additional computation, its overall cost remains comparable to many contemporary Transformer models. Since different backbones have varying hidden dimensions, the projection MLP to the encoder space incurs minor differences in FLOPs (within a few tenths of a GFLOP). For consistency, the FLOPs reported in table 7 assume a backbone hidden dimension of 768. Notably, when $s_{\min} = 4$, ReSAF reduces to a single MLP mapping backbone features to the encoder dimension.

Table 7: Computational cost.

| Component | GFLOPs |
|---|---|
| ClearCLIP | 17.35 |
| MMBPR | $15.72 \times (N + 1)$ |
| ReSAF | 5.78 |
| others | Negligible |

**Inference.** At test time, only the global branch is retained, resulting in a lightweight pipeline consisting of the backbone, one ReSAF module, and three encoder layers. Detailed efficiency statistics are provided in appendix J. Compared with the backbone alone, PSCL incurs only modest overhead and remains practical for deployment.

## 5 CONCLUSION

We introduce Part-level Semantic-guided Contrastive Learning (PSCL), a framework for Fine-Grained Visual Classification that jointly models part-level details and spatial relations for both rigid and non-rigid objects. PSCL employs a Part Localization Module (PLM) with ClearCLIP for semantically guided, interpretable part extraction, a Multi-scale Multi-part Branch Progressive Reasoning (MMBPR) module to fuse fine-grained and global features, and a Visual-Language Contrastive Learning module with Multi-grained Text Features (VLCL-MG) to align subcategories via intermediate-level semantics. Experiments on five FGVC benchmarks demonstrate PSCL's robust performance and strong generalization.

## 6 DISCUSSIONS

**Limitations.** PSCL has two primary limitations. First, it is fundamentally constrained by task definition. The framework is designed for fine-grained recognition within a single coarse-grained domain where semantic parts can be consistently defined. When multiple heterogeneous coarse-grained domains with incompatible morphological structures are involved (e.g., iNaturalist), constructing a unified part vocabulary becomes infeasible, requiring separate models for different domains. Second, PSCL relies on ClearCLIP for patch-level part localization. When objects or semantic parts are small, localization becomes sensitive to input resolution and downsampling. Moreover, ambiguous part boundaries may introduce noisy masks, requiring additional processing. A more precise open-vocabulary segmentation model could further improve robustness in such cases.

ETHICS STATEMENT

Our work focuses on fine-grained visual classification using publicly available datasets. No human subjects, personally identifiable information, or sensitive content are involved. All datasets employed, including AIR, CAR, NAB, DOG, and others, are used strictly for research purposes in accordance with their respective licenses. We acknowledge the potential societal impacts of deploying FGVC models, such as reinforcing biases present in the training data, and emphasize that PSCL should be applied responsibly, with consideration for fairness and ethical implications.

REPRODUCIBILITY STATEMENT

To ensure reproducibility, we provide detailed descriptions of all model components, including the Part Localization Module (PLM), Multi-scale Multi-part Branch Progressive Reasoning (MMBPR), and Visual-Language Contrastive Learning with Multi-grained Text Features (VLCL-MG). Hyperparameters, training procedures, and evaluation protocols are specified in the manuscript. We release the code, trained model checkpoints, and the dataset, enabling other researchers to reproduce our experiments under the same settings.

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

## A  THE USE OF LARGE LANGUAGE MODELS

In this work, we leverage large language models (LLMs), specifically ChatGPT-4o, to construct intermediate category hierarchies. Beyond this, LLMs are also employed to assist in code development and manuscript refinement. All outputs from the LLM are carefully verified to ensure accuracy. However, the intermediate category hierarchies may contain some errors due to their reliance on domain-specific expert knowledge, representing a potential source of noise in the experimental results.

## B  TRAINING AND HYPERPARAMETER SETTINGS FOR DIFFERENT BACKBONES

Table 8: Hyperparameter settings for different backbones.

| Hyperparameter Setting | ResNet-50 | ViT-B | Swin-B |
|---|---|---|---|
| Input resolution | $448 \times 448$ | $518 \times 518$ | $384 \times 384$ |
| Batch size | 16 | 16 | 16 |
| Weight decay | 0.01 | 0.01 | 0.01 |
| Optimizer | AdamW | AdamW | AdamW |
| Optimizer $\beta$ | (0.9, 0.95) | (0.9, 0.95) | (0.9, 0.95) |
| Optimizer $\epsilon$ | 1e$-$8 | 1e$-$8 | 1e$-$8 |
| Initial learning rate | 1e$-$4 | 1e$-$5 | 1e$-$5 |
| Learning rate schedule | Cosine annealing | Cosine annealing | Cosine annealing |
| Warm-up epochs | 10 | 10 | 10 |
| Epochs | 100 | 100 | 100 |
| Focusing parameter $\gamma$ | 4 | 4 | 4 |
| Smoothing noise factor $\epsilon_s$ | [0.4,0.3,0.2,0.1] | [0.4,0.3,0.2,0.1] | [0.4,0.3,0.2,0.1] |
| Multi-scale loss weight $\tilde{\epsilon}_s$ | [0.1,0.2,0.4,1.0] | [0.1,0.2,0.4,1.0] | [0.1,0.2,0.4,1.0] |

Table 9: Training time (hours) and peak VRAM (GB) for each model and dataset.

| Dataset | RN50 | | ViT-B | | Swin-B | |
|---|---|---|---|---|---|---|
| | Time (h) | VRAM (GB) | Time (h) | VRAM (GB) | Time (h) | VRAM (GB) |
| AIR | 3.7 | 18 | 5.1 | 22 | 4.8 | 23.8 |
| CAR | 3.8 | 15.1 | 5.4 | 19.1 | 5.1 | 21.6 |
| CUB | 3.0 | 16.5 | 4.2 | 20.6 | 3.9 | 22.7 |
| NAB | 12 | 16.5 | 16.6 | 20.6 | 15.4 | 22.7 |
| DOG | 5.5 | 15.1 | 8.0 | 19.1 | 7.5 | 21.6 |

Table 9 shows the training time and peak VRAM for each backbone and dataset. RN50 is generally faster and uses less memory than ViT-B and Swin-B, while larger datasets (e.g., NAB) require more time.

## C  PART TEXT FOR DIFFERENT DATASETS

| Dataset | Part Text |
|---|---|
| AIR | background of a plane, tail of a plane, logo of a plane, engine of a plane, landing gear of a plane, fuselage of a plane |
| CUB | background of a bird, head of a bird, foot of a bird, body of a bird, mouth of a bird |
| CAR | background of a car, head of a car, body of a car, back of a car |
| NAB | background of a bird, head of a bird, foot of a bird, body of a bird, mouth of a bird |
| DOG | background of a dog, head of a dog, foot of a dog, body of a dog |

# D OTHER RESULTS.

Table 10: Classification accuracy (%) on fine-grained datasets using different embedding/masking strategies.

| Method | ResNet50 | | | ViT-B | | |
|---|---|---|---|---|---|---|
| | CUB | Aircraft | Car | CUB | Aircraft | Car |
| Random text embeddings ($F_{\text{text}}$) | 88.93 | 94.75 | 95.38 | 92.04 | 95.45 | 96.36 |
| Random masking ($S(\mathbf{k})$) | 88.12 | 94.93 | 95.25 | 91.76 | 95.26 | 96.26 |
| Part text embeddings | 89.13 | 95.14 | 95.59 | 92.34 | 96.48 | 96.44 |

We conducted experiments in which either $F_{\text{text}}$ or $S(\mathbf{k})$ was randomized. Both random strategies can be viewed as mutually exclusive data augmentation methods based on random masking. However, Random Text Embeddings tend to occlude semantically similar regions, whereas Random Masking hides regions randomly. Our proposed PSCL architecture demonstrates considerable robustness: thanks to the MMBPR and VLCL-MG modules, the model can still learn to focus on relevant regions autonomously. Nevertheless, providing targeted human guidance could further improve the efficiency of this process.

Table 11: Effect of intermediate-category text annotations on NAB classification performance using ViT-B. Accuracy (%) is reported.

| Intermediate-Category Text | Accuracy (%) |
|---|---|
| Expert annotations (precise hierarchy) | **93.74** |
| Generated via ChatGPT-4o (semi-automatic) | 93.48 |
| Random-text control group | 92.83 |

We posit that the NAB dataset benefits substantially from its inherent, precise hierarchical category structure, resulting in a significant performance boost. Accordingly, we employed ChatGPT-4o in a semi-automatic manner to generate intermediate-category text annotations, while also creating a random-text control group, and conducted comparative experiments using the ViT-B backbone. The results demonstrate that accurate expert annotations effectively activate the VLCL-MG module, yet even the generated intermediate-category text can improve classification accuracy to a certain extent.

Table 12: Performance of different numbers of part texts on AIR and CUB datasets

| Dataset | Part text | N | Accuracy % |
|---|---|---|---|
| AIR | background of a plane, tail of a plane, Logo of a plane, engine of a plane, landing gear of a plane, fuselage of a plane | 6 | 95.14 |
| AIR | background of a plane, tail of a plane, head of a plane, fuselage of a plane | 4 | 94.74 |
| AIR | background of a plane, plane | 2 | 94.75 |
| AIR | plane | 1 | 94.66 |
| CUB | background of a bird, head of a bird, foot of a bird, body of a bird, mouth of a bird | 5 | 89.13 |
| CUB | background of a bird, head of a bird, body of a bird | 3 | 88.83 |
| CUB | background of a bird, bird | 2 | 88.47 |
| CUB | bird | 1 | 88.44 |

We conducted experiments to analyze the effect of the number of part texts ($N$). The results below suggest that 4–6 part texts offer a good balance between performance and complexity. Using even a single part text (e.g., "bird") still yields competitive results, as other effective components (like

progressive learning) contribute significantly. While using more parts increases computational cost, the performance gains diminish marginally. The experiments were performed using RN50.

# E    EXAMPLES OF SOME INTERMEDIATE CATEGORIES

Table 13: Intermediate classes for the AIR dataset

| Fine-grained | Intermediate-grained 1 | Intermediate-grained 2 |
|---|---|---|
| 737-900 | narrow-body airliner | twinjet |
| 747-100 | wide-body airliner | four-engined jet aircraft |
| A330-300 | wide-body airliner | twinjet |
| A340-200 | wide-body airliner | four-engined jet aircraft |
| Cessna 525 | business jet | twinjet |
| Challenger 600 | business jet | twinjet |
| DC-10 | wide-body airliner | trijet |
| DC-3 | cargo aircraft | twin-turboprop |
| Gulfstream V | business jet | twinjet |
| Hawk T1 | light aircraft | single-engine jet |
| Il-76 | cargo aircraft | four-engined jet aircraft |
| L-1011 | wide-body airliner | trijet |
| MD-11 | wide-body airliner | trijet |

Table 14: Intermediate classes for the CUB dataset

| Fine-grained | Intermediate-grained 1 | Intermediate-grained 2 |
|---|---|---|
| Frigatebird | Seabirds | Waterbirds |
| Gadwall | Ducks | Waterbirds |
| American Goldfinch | Finches | Songbirds |
| Boat-tailed Grackle | Grackles | Songbirds |
| American Crow | Crows | Corvids |
| Fish Crow | Crows | Corvids |
| Black-billed Cuckoo | Cuckoos | Songbirds |
| Rusty Blackbird | Blackbirds | Songbirds |
| Yellow-headed Blackbird | Blackbirds | Songbirds |
| Indigo Bunting | Buntings | Songbirds |

Table 15: Intermediate classes for the CAR dataset

| Fine-grained | Intermediate-grained 1 | Intermediate-grained 2 |
|---|---|---|
| Audi S4 Sedan 2007 | Sedan | Performance Vehicle |
| Audi TT RS Coupe 2012 | Coupe | Performance Vehicle |
| BMW ActiveHybrid 5 Sedan 2012 | Sedan | Hybrid Vehicle |
| BMW 1 Series Convertible 2012 | Convertible | Luxury Vehicle |
| BMW 1 Series Coupe 2012 | Coupe | Luxury Vehicle |
| Acura Integra Type R 2001 | Coupe | Performance Vehicle |
| Acura ZDX Hatchback 2012 | Hatchback | Luxury Vehicle |
| Aston Martin V8 Vantage Convertible 2012 | Convertible | Luxury Vehicle |
| Chrysler Crossfire Convertible 2008 | Convertible | Performance Vehicle |
| Chrysler PT Cruiser Convertible 2008 | Convertible | Family Car |
| Daewoo Nubira Wagon 2002 | Wagon | Family Car |

Table 16: Intermediate classes for the DOG dataset

| Fine-grained | Intermediate-grained 1 | Intermediate-grained 2 |
|---|---|---|
| Blenheim Spaniel | Sporting | Spaniel |
| Papillon | Toy | Toy-group |
| Toy Terrier | Toy | Terrier-toy |
| Rhodesian Ridgeback | Hound | Sighthound |
| Afghan Hound | Hound | Sighthound |
| Weimaraner | Sporting | Pointer |
| Staffordshire Bullterrier | Terrier | Bull-type |
| Cocker Spaniel | Sporting | Spaniel |
| Pug | Toy | Toy-group |
| Great Pyrenees | Working | Working-group |
| Irish Water Spaniel | Sporting | Spaniel |
| Kuvasz | Working | Working-group |
| Groenendael | Herding | Herding-group |

Table 17: Intermediate classes for the NAB dataset

| Fine-grained | Intermediate-grained 1 | Intermediate-grained 2 |
|---|---|---|
| Black-bellied Whistling-Duck | Black-bellied Whistling-Duck | Ducks, Geese, and Swans |
| Semipalmated Plover | Semipalmated Plover | Plovers, Sandpipers, and Allies |
| American White Pelican | American White Pelican | Pelicans, Herons, Ibises, and Allies |
| Killdeer | Killdeer | Plovers, Sandpipers, and Allies |
| Chimney Swift | Chimney Swift | Swifts and Hummingbirds |
| American Oystercatcher | American Oystercatcher | Plovers, Sandpipers, and Allies |
| Ross's Goose | Ross's Goose | Ducks, Geese, and Swans |
| Barn Owl | Barn Owl | Owls |
| Turkey Vulture | Turkey Vulture | Hawks, Kites, Eagles, and Allies |
| Brown Pelican | Brown Pelican | Pelicans, Herons, Ibises, and Allies |
| Scaled Quail | Scaled Quail | Grouse, Quail, and Allies |
| Rock Pigeon | Rock Pigeon | Pigeons and Doves |
| Black-necked Stilt | Black-necked Stilt | Plovers, Sandpipers, and Allies |

## F    LLM PROMPT

We use ChatGPT-4o with the following prompt (AIR example):

> "Please classify the following fine-grained categories based on visually discernible characteristics. Each category must belong to two distinct intermediate-grained categories. Output one line per category:
> [Fine-grained], [Intermediate-1], [Intermediate-2]:
> 707-320 727-200 737-200 ..."

## G    LLM NOISE-SENSITIVITY EXPERIMENTS

We additionally evaluated DeepSeek-R1 and Qwen-2.5-Max under greedy decoding. The results are summarized in table 18.

Table 18: Noise-sensitivity experiment results (accuracy %).

| Setting | Model | Accuracy |
|---|---|---|
| Baseline | ChatGPT-4o | 95.14 |
| Re-label using same intermediate-grained categories | DeepSeek-R1 | 95.08 |
| | Qwen-2.5-Max | 95.11 |
| Re-generate intermediate-grained categories | DeepSeek-R1 | 94.83 |
| | Qwen-2.5-Max | 95.21 |

Only DeepSeek-R1's regenerated taxonomy became overly coarse (e.g., engine type, wing type), but LLM variation overall shows robustness. All evaluated models outperform the version without VLCL-MG (94.54%).

## H    INPUT RESOLUTION OF OUR MODEL

For all RN50 models, the input resolution is consistent. For Swin-B, we adopt $384 \times 384$ as in HERBS, while TransIFC+ and CSQA-Net use $448 \times 448$. ViT-NeT uses $224 \times 224$.

The ViT-B model is somewhat special. After multiple trials, we selected ViT-B/14 with $518 \times 518$ resolution as the optimal choice. Some additional results for ViT-B models with different input resolutions are shown in table 19.

Table 19: Performance of ViT-B models under different input resolutions (accuracy %).

| Model | Resolution | AIR | CUB | DOG |
|---|---|---|---|---|
| ViT-B/14 | $518 \times 518$ | 96.48 | 92.34 | 92.27 |
| ViT-B/14 | $336 \times 336$ | 92.62 | 87.21 | 90.02 |
| ViT-B/16 | $448 \times 448$ | 94.31 | 92.23 | 91.01 |
| ViT-B/16 | $384 \times 384$ | 93.58 | 88.65 | 92.66 |

The performance of ViT-B on AIR improves significantly at higher resolutions. We attribute this to the increased resolution and smaller patch size, which allows the model to capture finer details of parts such as "logo" and "engine".

We note that reporting the best-performing resolutions might raise concerns regarding fairness. To address this, in *Comparison with Other Methods* we explicitly report both $448 \times 448$ and $518 \times 518$ settings for ViT-B. The $448 \times 448$ configuration ensures direct and fair comparison with existing methods, while the $518 \times 518$ setting reflects the commonly adopted optimal performance. Even when evaluated at the same resolution ($448 \times 448$), ViT-B models still achieve strong performance across datasets, demonstrating that the observed improvements are not due to resolution advantages or unfair comparisons.

## I ReSAF Operation Details

Additional ablation studies on ReSAF variants (with and without eq. (11) concatenation) are provided below, conducted on the AIR dataset with RN50 backbone:

Table 20: Ablation study of ReSAF variants on the AIR dataset

| Mechanism | w/o Eq.11 Concat (%) | w/ Eq.11 Concat (%) |
|---|---|---|
| SAE(Xu et al., 2025) | 93.67 | 93.79 |
| SAE + Positional Encoding | 94.33 | 94.43 |
| Prog. Cross-Attention | 94.60 | 94.99 |
| Prog. Flipped Key Cross-Attention | 94.42 | 94.78 |
| **ReSAF (Ours)** | 94.51 | **95.14** |

ReSAF leverages hierarchical feature roles: deep layers drive classification, while shallow layers assist. The flipped-key mechanism ensures complementary shallow information is captured in regions overlooked by deep features. Eq. 11 concatenation delays shallow-feature fusion, allowing the encoder and loss function to select the most useful features and preventing interference between shallow and deep layers. The experimental results above validate the effectiveness and reliability of ReSAF.

## J Inference Efficiency

We report the computational cost of PSCL across different backbones using a single NVIDIA RTX 4090 GPU. The encoder hidden dimension is set to 768, and the number of stages is fixed at 4 ($s_{\min=1}$). Table 21 summarizes per-image inference time, throughput, and peak VRAM for different batch sizes.

Table 21: Inference statistics. Time in ms/image; Throughput in images/s; PSCL: prediction head; Backb.: backbone; VRAM: peak GPU memory (GB).

| Batch | Model | Time (ms) | | Throughput (img/s) | | VRAM (GB) |
|---|---|---|---|---|---|---|
| | | PSCL | Backb. | PSCL | Backb. | |
| 1 | ViT-B | 17.26 | 6.39 | 57.94 | 156.59 | 4.54 |
| 1 | RN50 | 18.29 | 6.56 | 54.66 | 152.47 | 3.65 |
| 1 | Swin-B | 31.41 | 21.75 | 31.84 | 45.98 | 4.71 |
| 8 | ViT-B | 5.18 | 3.97 | 193.13 | 251.91 | 4.78 |
| 8 | RN50 | 3.11 | 1.38 | 321.89 | 724.25 | 3.93 |
| 8 | Swin-B | 5.01 | 3.45 | 199.79 | 289.70 | 5.04 |

## K Performance Using Only Last Stage Output and Inference Feature Selection

During inference, our model exclusively utilizes the global branch at the final stage. This choice is supported by empirical results and design considerations summarized as follows.

### 1. Role of Part-Level Branches and Inference Efficiency

Part-level branches function as regional prompts and feature refinement modules during training. However, the slight performance improvement they offer at inference time does not justify their computational cost. Therefore, we exclude these branches during deployment.

Comparisons between using only the global branch prediction $\mathbf{P}^{\mathrm{global}} = \mathbf{P}_{s=4,n'=0}$ and aggregating part-level predictions $\mathbf{P}^{\mathrm{sum}} = \sum_{n'=1}^{N+1} \mathbf{P}_{s=4,n'}$ are shown below:

Table 22: Comparison of inference strategies using only the global branch vs. summing part-level predictions. Accuracy (%) is reported for CUB, AIR, and CAR datasets across different backbones.

| Method | RN50 (CUB/AIR/CAR) | Swin-B (CUB/AIR/CAR) | ViT-B (CUB/AIR/CAR) |
|---|---|---|---|
| $\mathbf{P}^{\text{global}}$ | 89.13/95.14/95.59 | 93.01/95.32/95.54 | 92.34/96.48/96.44 |
| $\mathbf{P}^{\text{sum}}$ | 89.08/95.17/95.55 | 93.02/95.38/95.55 | 92.29/96.36/96.44 |

## 2. PROGRESSIVE CONFIDENCE ENHANCEMENT IN VLCL-MG

The VLCL-MG module introduces progressively strengthened confidence constraints across stages. As features propagate through the hierarchy, earlier-stage representations are processed and concatenated to later stages (eq. (11)), allowing the final stage to integrate comprehensive multi-stage information. To examine the effectiveness of different inference strategies, we compare:

$$\mathbf{P}^1 = \sum_{s=s_{\min}}^{4} \mathbf{P}_s, \qquad \mathbf{P}^2 = \sum_{s=s_{\min}}^{4} \tilde{\epsilon}_s \cdot \mathbf{P}_s, \qquad \mathbf{P}^3 = \mathbf{P}_{s=4,n'=0}.$$

Results on the CUB dataset validate the superiority of using only the final stage:

Table 23: Comparison of different inference strategies on the CUB dataset. Accuracy (%) is reported for each backbone.

| Method | RN50 | Swin-B | ViT-B |
|---|---|---|---|
| $\mathbf{P}^1$ | 88.40 | 92.71 | 90.95 |
| $\mathbf{P}^2$ | 88.95 | **93.09** | 92.12 |
| $\mathbf{P}^3$ | **89.13** | 93.01 | **92.34** |

Overall, relying solely on the final-stage prediction ($\mathbf{P}^3$) provides the best balance between accuracy and computational efficiency. While minor fluctuations may appear in certain individual cases, the final-stage strategy ($\mathbf{P}^3$) remains the most reliable and effective option when considering overall performance and practical deployment constraints.

## L ADDITIONAL VISUALIZATIONS OF PART SCORES

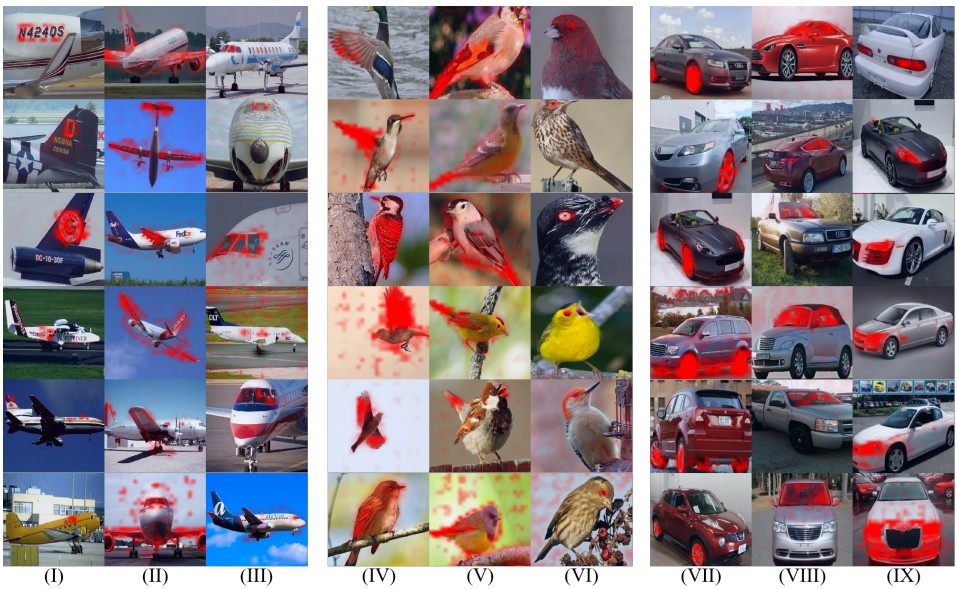

(I)    (II)    (III)    (IV)    (V)    (VI)    (VII)    (VIII)    (IX)

Figure 7: Additional visualizations of part scores. PLM uses the following textual prompts: (I) logo; (II) wing; (III) windows; (IV) wing; (V) tail; (VI) eyes; (VII) wheels; (VIII) windows; (IX) headlights.

