# OpenReview forum: "Part-level Semantic-guided Contrastive Learning for Fine-grained Visual Classification"
_ICLR.cc/2026/Conference — ICLR 2026 Poster_

### Official Review · Reviewer_fUYe · 2025-10-14

**Soundness:** 3
**Presentation:** 2
**Contribution:** 2
**Rating:** 4
**Confidence:** 5

**Summary:**

This paper focuses on the task of fine-grained visual classification and proposes the part-level semantic-guided contrastive learning approach. It consists of three modules, i.e., part localization module (PLM), multi-scale multi-part branch progressive reasoning (MMBPR), and visual-language contrastive learning based on multi-grained text features (VLCL-MG). Experiments are conducted on five fine-grained image datasets.

**Strengths:**

- The writing is easy to follow.
- Experiments are conducted on five datasets with three different backbones.

**Weaknesses:**

- The core idea of PLM for localizing part regions appears similar to [a]. PLM leverages CLIP to select N part masks, but the paper does not clarify how N is chosen or how performance changes with different numbers of parts.
- The paper claims that s_min can be adjusted based on task requirements; however, this is unclear. What is its influence? And the idea of multi-scale features has been widely used.
- MMBPR works simply by concatenating the class token with multi-scale multi-part features, then processing the result with LayerNorm (LN), multi-head self-attention (MHSA), and an MLP. As shown in Table 6, the gain from MMBPR (line 3 vs. line 2) is marginal, leaving its effectiveness unconvincing.
- Although VLCL-MG leverages label smoothing, focal loss, and auxiliary knowledge from intermediate categories, the observed improvements are marginal, as evidenced by Table 6 (line 5 versus line 3).
- Table 2 indicates that the proposed method yields only marginal gains over SOTA baselines, particularly on CUB, CAR, and AIR. An additional evaluation on iNaturalist is recommended. Besides, this method utilizes extra information, including CLIP, textual information, and intermediate categories, which are not used in SOTA methods.
- Since image resolution has a direct impact on performance, the paper should specify the resolutions used in all compared methods to ensure a fair comparison.
- Figure 1 highlights the spatial structures of non-rigid vs. rigid objects, but the method does not explicitly model them. Incorporating spatial structure could be promising.
- There are issues with grammar and presentation. On line 250, “TWe” appears to be a typo. In addition, Figure 2 may be overly complex and hard to follow.


[a] Part-guided Relational Transformers for Fine-grained Visual Recognition

**Questions:**

My questions are shown in weaknesses.

---

> ### Author Response · Authors · 2025-11-24
>
> We sincerely thank the reviewer for the constructive and detailed feedback. Below we respond to each concern point-by-point and clarify the design motivations, empirical findings, and contributions of our work.
>
> ---
>
> **Q1: On the novelty of PLM and its difference from PART [1], the choice of N, and its influence.**
>
> **A1:** We appreciate the reviewer’s observation regarding the similarity between PLM and prior part-based methods. Indeed, following early works such as PART [1], multi-part-branch structures have become a common paradigm in state-of-the-art FGVC systems. However, existing approaches share a structural limitation: the same feature extractor is responsible both for generating spatially discriminative regions and for performing classification. As a consequence, the identified “parts” are not semantically grounded and are often just those regions most beneficial for classification.
>
> In PART [1], for example, the model relies on class activation maps produced during gradient descent and selects regions using saliency-based ranking. Such a mechanism does not ensure that each branch consistently captures a specific part across samples. Particularly in the presence of occlusion, pose variation, or deformation—common in non-rigid categories—the same branch may attend to entirely different visual components across images, causing high redundancy and unstable learning across part branches.
>
> **Our PLM fundamentally differs in this respect.**
> By leveraging text prompts and ClearCLIP-based part localization, PLM enables semantically grounded, controllable, and interpretable part selection, where each branch is explicitly associated with a specific part. Importantly, PLM decouples region selection from visual representation. Even if a part is occluded, the mask-based formulation of Eq. (6) temporarily deactivates its branch by preventing gradients from flowing, thereby maintaining stable learning. During inference, only the global branch is used; missing part branches do not affect prediction.
>
> **On choosing the number of parts (N):**
> The part texts are predefined based on the coarse-grained category and serve to guide ClearCLIP. We evaluated the influence of N on AIR and CUB using RN50, obtaining:
>
> **AIR (RN50):**
>
> | Part text| N | Accuracy % |
> | -| - | --|
> | background of a plane, tail of a plane, Logo of a plane, engine of a plane, landing gear of a plane, fuselage of a plane | 6 | 95.14  |
> | background of a plane, tail of a plane, head of a plane, fuselage of a plane | 4 | 94.74  |
> | background of a plane, plane | 2 | 94.75  |
> | plane| 1 | 94.66  |
>
> **CUB (RN50):**
>
> | Part text | N | Accuracy % |
> | -| - | -|
> | background of a bird, head of a bird, foot of a bird, body of a bird, mouth of a bird | 5 | 89.13  |
> | background of a bird, head of a bird, body of a bird  | 3 | 88.83  |
> | background of a bird, bird| 2 | 88.47  |
> | bird  | 1 | 88.44  |
>
> We found that 4–6 part texts strike the best balance: as N increases, computational cost increases significantly while marginal gains diminish.
>
> ```
> [1] Part-guided Relational Transformers for Fine-grained Visual Recognition
> ```
>
> ---
>
> **Q2: On the meaning of $ s_{\min} $, its influence, and the role of multi-scale features**
>
> **A2:** The value $ s_{\min} $ determines whether shallow-stage features from the backbone are included. As demonstrated in works like PMG [2], FGVC tasks often require fine texture or low-level detail, depending on dataset characteristics and backbone inductive bias. When $ s_{\min} < 4 $, PLM and MMBPR incorporate intermediate-level features to provide additional detail; when shallow features are unnecessary or hardware constraints apply, a larger $ s_{\min} $ can remove those stages for speed.
>
> Comparing lines 2 and 3 in Table 6, we observe that shallow features contribute differently across backbones: with Swin-B, both AIR and CAR benefit from the inclusion of shallow stages, whereas with ViT-B, the CUB dataset gains the most from retaining shallow features.
>
> Thus, $ s_{\min} $ is intentionally adjustable. Multi-scale features are not the novelty of our work but a necessary part of a complete FGVC system; the key contribution lies in how PLM and MMBPR mitigate redundancy between scales.
>
> ```
> [2] Fine-grained visual classification via progressive multi-granularity training of jigsaw patches.
> ```

---

> > ### Author Response · Authors · 2025-11-24
> >
> > **Q3: On the effectiveness of MMBPR and its seemingly marginal gains**
> >
> > **A3:** MMBPR is not designed as a standalone enhancement module; rather, it acts as a structural bridge between PLM and VLCL-MG, enabling progressive reasoning and preventing low-level features from interfering with high-level feature refinement.
> >
> > The design choices are motivated by:
> >
> > The input to MMBPR is derived from the output of ReSAF, which carefully manages multi-scale feature interactions: ReSAF structures feature contributions based on their importance during classification: final-layer features dominate, while shallow features provide auxiliary cues. To ensure that complementary information from earlier stages is not overlooked, ReSAF uses a flipped-key mechanism to focus on regions that deep features fail to attend to. At the same time, to avoid interference between shallow and deep features—since aggressive multi-scale fusion can harm deeper-stage representations—the concatenation (Eq. 11) allows the encoder and loss function to select the most useful features automatically.
> >
> > To demonstrate the impact of merging or removing components, additional results (AIR, RN50) show:
> >
> > | Mechanism| w/o Eq.11 (%) | w/ Eq.11 (%) |
> > | --------------------------------------- | ------------- | ------------ |
> > | SAE [3]| 93.67| 93.79|
> > | SAE + Positional Encoding| 94.33| 94.43|
> > | Progressive Cross-Attention| 94.60| 94.99|
> > | Progressive Flipped Key Cross-Attention | 94.42| 94.78|
> > | **ReSAF (ours)**| **94.51** | **95.14**|
> >
> > These experiments show that naïve multi-scale fusion can degrade performance, while MMBPR’s design ensures stability. Gains appear small in some cases because deep features dominate the task (Table 6, line 2); however, MMBPR improves robustness across the vast majority of configurations. Since it is often difficult to determine whether shallow features are effective, using MMBPR is recommended to ensure their contribution is properly leveraged.
> >
> > ```
> > [3] Context-Semantic Quality Awareness Network for Fine-Grained Visual Categorization.
> > ```
> >
> > ---
> >
> > **Q4: On the marginal improvement of VLCL-MG**
> >
> > **A4:** The reviewer’s comment (Table 6, line 5 vs. line 3) is correct that combined gains diminish. This is expected, as PLM and VLCL-MG target highly correlated deficiencies inherent to FGVC:
> >
> > * PLM strengthens part-level feature extraction.
> > * VLCL-MG provides semantically coherent cluster centers derived from intermediate categories.
> >
> > Fine-grained categories that share intermediate categories—e.g., similar bird taxa or similar car types—often share similar parts. Consequently, the two modules address overlapping challenges. When comparing line 1 (baseline) and line 4 (only VLCL-MG), the VLCL-MG module offers substantial gains, demonstrating its independent impact.
> >
> > ---
> >
> > **Q5: On additional evaluation on iNaturalist and the use of extra information**
> >
> > **A5:** **(1)** Our framework is designed for fine-grained classification within a single coarse-grained domain. In contrast, iNaturalist contains 13 heterogeneous super-categories, each with distinct semantic parts and hierarchical structures, making it impossible to define a unified set of part texts or intermediate categories across all groups.
> >
> > Even under this constraint, we conducted a simplified evaluation with:
> >
> > * Only “object” and “background” part texts,
> > * Intermediate categories restricted to *(super-category, fine-grained class)*,
> > * ViT-B as the backbone,
> > * All other settings kept identical.
> >
> > **Result: 74.6% on iNat 2017**, outperforming TransFG (71.7) but below MDCM (79.8).
> > This demonstrates that PSCL generalizes reasonably well even when deprived of the richer semantic cues it is explicitly designed to leverage.
> >
> > **(2) Regarding the use of “extra information”:**
> > All additional information—such as part texts or intermediate categories—is provided once per class, rather than per image. Crucially, this information is task-dependent but data-independent: it can be obtained as long as the specific task (i.e., the super-class and its corresponding fine-grained categories) is known, without access to the actual input images. This makes the approach practical for real-world FGVC scenarios. Table 9 demonstrates that even semi-automated LLM-generated texts achieve competitive performance.
> >
> > One potential concern is that leveraging additional semantic information might introduce biases or raise fairness issues. However, since the information is class-level and task-specific rather than instance-specific, it does not directly depend on individual input data. Therefore, it is less likely to exacerbate fairness problems that arise from dataset-specific biases. To our knowledge, our method is the first to inject such rich semantic information at this cost, and there exists no prior work suitable for a fair comparison.

---

> > > ### Author Response · Authors · 2025-11-24
> > >
> > > **Q6: On image resolution and fairness**
> > >
> > > **A6:** All resolutions used in comparison are reported in Implementation Details.
> > >
> > > * RN50: 448×448 for all methods
> > > * Swin-B: 384×384 for ours and HERBS; 448×448 for TransIFC+ and CSQA-Net
> > > * ViT-B: 518×518 (best-performing resolution)
> > >
> > > | Model | Resolution | AIR | CUB | DOG |
> > > |---------|--------|-----|-----|-----|
> > > | ViT-B/14 | 518×518 | 96.48 | 92.34 | 92.27 |
> > > | ViT-B/14 | 336×336 | 92.62 | 87.21 | 90.02 |
> > > | ViT-B/16 | 448×448 | 94.31 | 92.23 | 91.01 |
> > > | ViT-B/16 | 384×384 | 93.58 | 88.65 | 92.66 |
> > >
> > > Additional experiments show that ViT-B performance on AIR is particularly sensitive to resolution, likely due to improved perception of fine details (e.g., “logo”, “engine”). We will include these results in the Appendix H to clarify potential concerns regarding fairness. Even at the same resolution, ViT-B models still demonstrate strong performance, indicating that the observed improvements are not due to an unfair comparison.
> > >
> > > ---
> > >
> > > **Q7: On spatial structure modeling**
> > >
> > > **A7:** We agree with the reviewer that explicit spatial modeling is promising. We intentionally avoid explicit modeling in part branches because:
> > >
> > > 1. Inference costs would increase substantially if part branches must run during inference.
> > > 2. Our design philosophy aligns with prior FGVC works—part branches serve only to refine the global branch.
> > >
> > > The global branch, through repeated training with shared positional encodings, gradually acquires implicit spatial modeling capability. Spatial information flows from part branches back to the backbone via gradients, enabling the global branch to internalize spatial relationships after sufficient training.
> > >
> > > ---
> > >
> > > **Q8: On grammar and figure complexity**
> > >
> > > **A8:** We thank the reviewer for identifying issues such as the typo “TWe” on line 250. These have been corrected. We have updated the textual description to facilitate a better understanding of Figure 2, and the simplification of the figure is currently in progress.

---

> > ### Comment · Reviewer_fUYe · 2025-11-26
> >
> > I appreciate the authors’ rebuttal; however, I still have several concerns.
> > - From A1, we can see that the performance gains from using parts are rather modest, i.e., 0.48 and 0.64 on Air and CUB, respectively, and are reported under different part numbers. This is somewhat at odds with the paper’s main claimed contribution, which is the proposed part design.
> > - The additional results on iNaturalist further highlight the limitations of the proposed method when evaluated on large-scale validation data.
> > - For image resolution, I suggest that the authors also report the resolutions used in the compared methods to ensure a fair comparison.

---

> > > ### Author Response · Authors · 2025-12-04
> > >
> > > We thank the reviewer for the thoughtful comments and the opportunity to further clarify our contributions and experimental design.
> > >
> > > ---
> > >
> > > **1. On the magnitude of performance gains from part usage:**
> > >
> > > The reviewer is correct that the incremental gains reported in Table A1 may appear modest. This is primarily because the results were obtained using the final version of our codebase under different numbers of part prompts. As noted previously in Q4–A4, the diminishing marginal gains are expected.
> > >
> > > Moreover, although the absolute improvements appear small, they are meaningful given the already strong baselines and the intrinsic difficulty of distinguishing extremely similar sub-categories in FGVC.
> > >
> > > To provide further evidence, we re-ran the part-number ablation experiments on models with MMBPR and VLCL-MG removed:
> > >
> > > **AIR (RN50):**
> > > |Part text|N|Accuracy (%)|
> > > |-|-|-|
> > > | background of a plane, tail of a plane, Logo of a plane, engine of a plane, landing gear of a plane, fuselage of a plane | 6 | 94.54 |
> > > | background of a plane, tail of a plane, head of a plane, fuselage of a plane | 4 | 93.67 |
> > > | background of a plane, plane | 2 | 93.16 |
> > > | plane | 1 | 92.80 |
> > >
> > > **CUB (RN50):**
> > > |Part text|N|Accuracy (%)|
> > > |-|-|-|
> > > | background of a bird, head of a bird, foot of a bird, body of a bird, mouth of a bird | 5 | 88.82 |
> > > | background of a bird, head of a bird, body of a bird | 3 | 87.21 |
> > > | background of a bird, bird | 2 | 87.07 |
> > > | bird | 1 | 86.45 |
> > >
> > > We emphasize that PLM introduces hidden-dimension misalignment and mask-induced sparsity issues, making it infeasible to completely remove MMBPR; therefore, part of the additional gains above baseline originate from these remaining components and their associated parameters.
> > >
> > > Finally, our main contribution is not merely increasing performance through part count, but **reducing feature redundancy across branches and establishing a unified spatial-awareness mechanism applicable to both rigid and non-rigid objects**. These benefits primarily affect **hard examples**, whereas a large proportion of FGVC samples are easy; this imbalance causes the improvements on hard cases to be visually “diluted” when averaged across the full dataset.
> > >
> > > ---
> > >
> > > **2. On the applicability to large-scale datasets such as iNaturalist**
> > >
> > > We agree that our current model has certain limitations, but these limitations do not stem from dataset scale. Rather, they arise from the **task definition**.
> > >
> > > Our framework is designed for **fine-grained classification within a single coarse-grained domain**, where semantic parts and intermediate categories can be defined consistently. In contrast, **iNaturalist contains 13 highly heterogeneous super-categories**, each characterized by distinct morphological structures and incompatible semantic parts. This makes it impossible to construct a unified set of part texts or shared intermediate categories across all groups.
> > >
> > > This mismatch reflects a difference in task formulation, not a failure of scalability. Indeed, a meaningful application of our model to iNaturalist would require **training separate models for each heterogeneous super-category**. Due to time constraints, we provided only a limited preliminary evaluation in Q5–A5, where only a subset of the architectural components were applicable. Even so, the model achieved reasonable accuracy, though not SOTA.
> > >
> > > Therefore, we do not believe these results indicate that our approach fails on large-scale data. Rather, our design is intentionally **specialized for domain-focused fine-grained recognition**, much like consulting a domain expert: asking a mycology specialist to identify a bird species is inherently suboptimal compared to consulting a trained ornithologist.
> > >
> > > ---
> > >
> > > **3. On reporting the image resolutions of compared methods**
> > >
> > > We appreciate the reviewer’s suggestion. The input resolutions used in our experiments are already provided in **Section 4.1 (Experimental Setup)** and **Appendix H**.
> > >
> > > In Section 4.2, we report the best performance under the settings defined in Section 4.1. While earlier FGVC works relied heavily on reporting input resolution for fairness, recent approaches frequently use overlapping cropping strategies, region-of-interest enlargement. These techniques introduce additional computational overhead, such that **input resolution is no longer a reliable or sufficient fairness metric**.
> > >
> > > Therefore, we instead provide:
> > >
> > > *-* **Full computational cost analysis in Section 4.3 (Inference Efficiency)**
> > > *-* **Detailed resolution settings and result in Appendix H**
> > >
> > > This allows readers to better understand both the actual computational requirements and the practical performance of the evaluated models.

---

### Official Review · Reviewer_UrRj · 2025-10-23

**Soundness:** 3
**Presentation:** 3
**Contribution:** 3
**Rating:** 6
**Confidence:** 4

**Summary:**

This paper proposes a new FGVC framework that leverages text-guided part localization, multi-level class embedding, and multi-grained vision-language contrastive learning. This framework is compatible with different backbones and achieves superior performance over baseline methods across benchmarks.

**Strengths:**

1. The motivation for overcoming the limitation of the common part-level interaction-based methods, such as CAP, is reasonable and well demonstrated in Figure 1 and supported by Figure 4.

2. The proposed method is compatible with different pretrained backbones and achieves consistent improvement.

3. The idea that leverages a single VLM to localize parts and conduct contrastive learning is novel.

4. The code is provided to ensure reproducibility.

**Weaknesses:**

1. The writing lacks some important details. For example, how to apply a multi-stage module on the single-scale ViT? Does ReSAF only work between f_{1,2,3} and f_4? Does it also work between f_{1,2} and f_3?

2. The explanation of Eq. 5 is confusing. How does this min-max operation perform in detail?

3. There is a typo in L249-250: "TWe".

4. The computation cost needs to be compared with baseline models, such as TransFG. It would be better to add at least one baseline model for different backbones.

5. What is the performance if only the last stage output is leveraged during training? Which feature is used for final categorization during inference?

**Questions:**

My questions are listed above in the weaknesses part.

---

> ### Author Response · Authors · 2025-11-24
>
> We thank the reviewers for their insightful comments and constructive feedback. Below, we address each point raised in the review.
>
> **Q1: Lack of Important Details in Writing (e.g., Multi-stage Module on Single-scale ViT, ReSAF Operation).**
>
> **A1:**  We appreciate the reviewer's feedback regarding the clarity of our methodological details. Here, we provide clarifications on the multi-scale feature handling in ViT and the ReSAF module.
>
> 1.  **Multi-level Feature Extraction in ViT:**  ViT-B consists of 12 Transformer encoder blocks. We extract features from specific layers (layers 4, 7, and 10) as shallow-level features, and the final layer output as deep-level features. After removing the [cls] token, we retain patch-level feature vectors to form multi-level features. As the reviewer rightly pointed out, ViT is inherently a single-scale architecture. Consequently, the multi-scale operations described in our model manifest as multi-level feature processing when applied to ViT. We acknowledge this lack of clarity in the original manuscript and have explicitly clarified this point in the first paragraph of Section 3 (Approach) in the revised version. This clarification does not affect the validity of our experimental results or conclusions. The MMBPR module remains effective on ViT by progressively refining feature representations from shallow to deep layers.
>
> 2.  **ReSAF Operation Details:**
>    - ReSAF is designed to mitigate redundancy by enabling cross-level attention. Specifically, when $s_{\text{min}} = 1$, ReSAF performs flipped key-vector cross-attention between each shallow-level feature ($f_1$, $f_2$, $f_3$) and the deep-level feature ($f_4$), followed by self-attention on $f_4$. All features share learnable positional encoding. For $s_{\text{min}} > 1$, the same operation applies between shallow features and $f_4$. ReSAF does not operate between shallow features (e.g., $f_1$ and $f_2$).
>    - ReSAF's design leverages hierarchical feature roles: final layers lead classification; shallow layers assist. A reversed-key mechanism directs attention to deep-feature-omitted areas. Avoiding shallow-layer interactions prevents attention overlap. Shallow-feature fusion is delayed through Eq.11 concatenation, handled by the encoder, and optimized via loss function.
>    - Additional ablation studies on ReSAF variants (with and without Eq.11 concatenation) are provided below, conducted on the AIR dataset with RN50 backbone:
>
> | Mechanism | w/o Eq.11 Concat (%) | w/ Eq.11 Concat (%) |
> |-----------|----------------------|---------------------|
> | Scale-aware Enhancement (SAE) [1] | 93.67 | 93.79 |
> | SAE + Positional Encoding | 94.33 | 94.43 |
> | Prog. Cross-Attention | 94.60 | 94.99 |
> | Prog. Flipped Key Cross-Attention | 94.42 | 94.78 |
> | **ReSAF (Ours)** | 94.51 | **95.14** |
>
>   - These results validate ReSAF's effectiveness.
>
> ```
> [1] Context-Semantic Quality Awareness Network for Fine-Grained Visual Categorization.
> ```
>
> **Q2: Confusing Explanation of Equation (5).**
>
> **A2:**  Equation 5 represents a morphological closing operation in image processing, which consists of dilation followed by erosion.
>
> 1.  **Explanation and Significance of the Original Equation:** The $\max$ operation indicates that the value of a region in $\mathbf{M}$ is determined by the maximum value within the kernel window $\mathcal{K}$ surrounding the corresponding region in $\mathbf{S}$. This corresponds to a dilation operation on the binarized image, while the $\min$ operation performs the opposite effect. In our framework, the primary purpose of this operation is to fill gaps in the masks caused by noise, occlusion, or overlapping parts. For instance, in the part masks of an aircraft, the logo mask is often located on the fuselage or tail, which may create blank areas within the masks of those parts. Applying a closing operation helps fill these gaps, enabling the corresponding branch to achieve better feature representation.
>
> 2. **Revision for Enhanced Clarity:** We acknowledge that the original explanation of Equation 5 may have been unclear. In the revised version, this section has been updated as follows:
> "
> $
> \mathbf{M} = (S \oplus \mathcal{K}) \ominus \mathcal{K},
> $
> where $\mathcal{K}$ denotes the structuring element (kernel), instantiated as a $3 \times 3$ kernel in our implementation; $\oplus$ represents the morphological dilation operator; and $\ominus$ denotes the erosion operator.
> "

---

> > ### Author Response · Authors · 2025-11-24
> >
> > **Q3: Typo in "TWe" (L249-250).**
> >
> > **A3:** We thank the reviewer for catching this typo. It has been corrected to "We" in the updated manuscript.
> >
> >
> > **Q4: Computation Cost Comparison with Baseline Models.**
> >
> > **A4:** During inference, our model aligns with early multi-branch algorithms in this field, such as [2], by utilizing only the global branch for prediction. Within our architecture, the part-level branches and ClearCLIP module can be entirely removed. Additional training and inference data can be found in Section 4.4 Training and Inference Efficiency of the revised version, as well as in Appendix B.
> >
> > 1.  **Inference Time and Computational Efficiency:** When processing four stages, the computational overhead comprises solely the backbone network, one ReSAF module, and three Encoder Layers. Specifically, under the configuration with an encoder hidden dimension of 768 and excluding image I/O operations, the inference speed for four stages is summarized in the table below. We directly compare against the original backbone models, with all experiments conducted on a single NVIDIA RTX 4090 GPU.
> >
> > **Per-image Inference Time (ms):**
> > | Batch Size | Model  | PSCL (ms) | Original Backbone (ms) |
> > |-|-|-|-|
> > | 1| ViT-B  | 17.26| 6.39|
> > | 1| RN50   | 18.29| 6.56|
> > | 1| Swin-B | 31.41| 21.75|
> > | 8| ViT-B  | 5.18| 3.97|
> > | 8| RN50   | 3.11| 1.38|
> > | 8| Swin-B | 5.01| 3.45|
> >
> > **Inference Throughput (images/second):**
> > | Batch Size | Model  | PSCL (it/s) | Original Backbone (it/s) |
> > |-|-|-|-|
> > | 1| ViT-B  | 57.94| 156.59|
> > | 1| RN50   | 54.66| 152.47|
> > | 1| Swin-B | 31.84| 45.98 |
> > | 8| ViT-B  | 193.13| 251.91|
> > | 8| RN50   | 321.89| 724.25|
> > | 8| Swin-B | 199.79| 289.70|
> >
> > 1.  **Revision for Enhanced Clarity:** We acknowledge that the original description of the inference process in our manuscript lacked sufficient elaboration. This aspect has been comprehensively addressed in the revised version with the following enhanced explanation:
> > "
> > At inference time, the prediction from the final stage is utilized, and the inference strategy relies solely on the global branch:
> > $
> >     \mathbf{P}^{\text{global}}=\mathbf{P} _ {s=4,\, n'=0},
> > $
> > in which ClearCLIP and redundant part-level branches are removed during inference, thereby enabling substantially faster computation.
> > "
> >
> > ```
> > [2] Zhao et al., "Part-guided Relational Transformers for Fine-grained Visual Recognition," IEEE TIP 2021.
> > ```
> >
> > **Q5: Performance Using Only Last Stage Output and Inference Feature Selection.**
> >
> > **A5:** Our model exclusively utilizes the global branch at the final stage during inference. This design choice is underpinned by empirical evaluations and theoretical considerations detailed below.
> >
> > 1.  **Role of Part-Level Branches and Inference Efficiency:** The part-level branches serve solely as regional prompts and feature refinement mechanisms during training. During inference, however, their marginal performance gain fails to justify the associated computational overhead, leading to their exclusion in the final deployment. The comparative results between using only the global branch prediction $\mathbf{P}^{\text{global}} = \mathbf{P}_{s=4, n'=0}$ and aggregating part-level branch predictions $\mathbf{P}^{\text{sum}} = \sum _ {n'=1}^{N+1} \mathbf{P} _ {s=4, n'}$ are summarized below:
> >
> > | Method| RN50 (CUB/AIR/CAR) | Swin-B (CUB/AIR/CAR) | ViT-B (CUB/AIR/CAR) |
> > |-|-|-|-|
> > | $\mathbf{P}^{\text{global}}$ | 89.13/95.14/95.59  | 93.01/95.32/95.54   | 92.34/96.48/96.44   |
> > | $\mathbf{P}^{\text{sum}}$    | 89.08/95.17/95.55  | 93.02/95.38/95.55   | 92.29/96.36/96.44   |
> >
> > 1.  **Progressive Confidence Enhancement in VLCL-MG.**
> > - Within the VLCL-MG module, training incorporates progressively increasing confidence constraints across stages. As features advance through stages, confidence is enhanced via hierarchical processing. Crucially, features from preceding stages are processed and concatenated to subsequent ones (Eq. 11), ensuring the final stage encapsulates comprehensive information.
> >
> > - To validate this, we compared three inference strategies:
> >   $\mathbf{P}^1 = \sum_{s=s _ {\min}}^{4} \mathbf{P} _ s$ (direct summation)
> >
> >   $\mathbf{P}^2 = \sum_{s=s _ {\min}}^{4} \tilde{\epsilon} _ s \cdot \mathbf{P} _ s$ (weighted summation)
> >
> >   $\mathbf{P}^3 = \mathbf{P} _ {s=4, n'=0}$ (final stage only)
> >
> > Results on the CUB dataset demonstrate the efficacy of $\mathbf{P}^3$:
> >
> > | Method| RN50  | Swin-B | ViT-B |
> > |-|-|-|-|
> > | $\mathbf{P}^1$ | 88.40 | 92.71  | 90.95  |
> > | $\mathbf{P}^2$ | 88.95 | 93.09  | 92.12  |
> > | $\mathbf{P}^3$ | **89.13** | **93.01** | **92.34** |
> >
> > - The final-stage strategy ($\mathbf{P}^3$) achieves optimal balance between accuracy and computational efficiency.

---

> > > ### Comment · Reviewer_UrRj · 2025-11-26
> > > **Post Rebuttal Responses by Reviewer**
> > >
> > > Thank you for the detailed responses. A minor bug is that in your A5, the $\mathbf{P}^3$ with Swin-B achieves 93.01, which is not the best score.
> > >
> > > In general, I appreciated the revised version as it is much better to help understand and re-implement the method. The detailed pipeline of proposed modules and the final structure for inference are clear.
> > >
> > > I have three more suggestions:
> > >
> > > (1) It would be better to distinguish the training structure and inference structure in **Figure 2**.
> > >
> > > (2) Please include the additional experiments in the revised manuscripts, as I could not find the discussion of **Q5-A5** in the revised version. This would help better understand why the final structure only uses the global branch.
> > >
> > > (3) It would be better to provide more visualization results in **Figure 4**, as it is the direct evidence to show the effectiveness of the vision-language part-level alignment.
> > >
> > > After reviewing all other reviews and responses, I support the paper for clear acceptance. Good luck!

---

> > > > ### Author Response · Authors · 2025-12-04
> > > >
> > > > Thank you very much for the constructive suggestions and for your positive assessment of our revised manuscript. Please find our responses below. We sincerely appreciate your careful reading and helpful comments.
> > > >
> > > > ---
> > > >
> > > > **1. Regarding the 93.01 result in A5:**
> > > >    Thank you for pointing out the minor inconsistency. We acknowledge that the 93.01 result for Swin-B was mistakenly emphasized. This value was newly added in **Appendix J** during the revision, and we have now corrected its formatting to avoid misunderstanding. Importantly, this adjustment does not affect our overall conclusion:
> > > >    *“While minor fluctuations may appear in certain individual cases, the final-stage strategy $(\mathbf{P}^3)$ remains the most reliable and effective option when considering overall performance and practical deployment constraints.”*
> > > >
> > > > ---
> > > >
> > > > **2. Distinguishing training vs. inference structures in Figure 2:**
> > > >    We have further refined Figure 2 in the revised version. The improvements include:
> > > >    - Explicitly distinguishing the training pipeline from the inference pipeline;
> > > >    - Removing redundant or potentially confusing symbols;
> > > >    - Streamlining several expressions to enhance visual clarity and conceptual flow.
> > > >
> > > > ---
> > > >
> > > > **3. Including additional experiments (Q5–A5):**
> > > >    The analysis and discussions related to Q5–A5 have been consolidated and included in **Appendix J**, along with the corrected numerical entry mentioned above. These additions help clarify why the final structure adopts only the global branch at inference.
> > > >
> > > > ---
> > > >
> > > > **4. Enriching the visualization results:**
> > > >    We have more visualization results corresponding to Figure 4 in **Appendix K**. These additional examples provide further direct evidence of the effectiveness of our vision–language part-level alignment strategy.
> > > >
> > > > ---
> > > >
> > > > Thank you again for your thoughtful feedback and your support for clear acceptance. We greatly appreciate your time and insights.

---

### Official Review · Reviewer_WQPm · 2025-10-27

**Soundness:** 3
**Presentation:** 3
**Contribution:** 2
**Rating:** 4
**Confidence:** 5

**Summary:**

This paper proposed Part-level Semantic-guided Contrastive Learning to capture both part-level detail and spatial relational features for FGVC task.
Authors combine vision–language alignment, part reasoning, and progressive confidence modeling into an unified framework.

**Strengths:**

Consistent SOTA results on 5 datasets and across CNN/Transformer backbones.

The CLIP-based masks yield interpretable part attention for recognition.

**Weaknesses:**

The biggest problem with this paper is that the design is overly complex, while the improvement is relatively limited.

Is the number of part texts predetermined? How does the method adapt to different tasks, especially in cases where the number of part texts varies across categories within the same classification task?

Can large language models accurately describe the differences between categories? Coarse-grained differences are easy to express, such as head color or beak shape. However, some subtle differences are difficult to localize and describe using language.

**Questions:**

Please refer to weakness.

---

> ### Author Response · Authors · 2025-11-24
>
> We thank the reviewers for their insightful comments and constructive feedback. We address the concerns raised below.
>
> **Q1: The design is overly complex, while the improvement is relatively limited.**
>
> **A1:** We appreciate the reviewer's feedback regarding the complexity. We would like to clarify that each component in our framework is designed to address specific challenges in Fine-Grained Visual Classification (FGVC).
>
> 1.  **Purpose of Components:** Each component addresses a specific limitation. The **PLM** decouples region proposal from feature learning via text guidance. The **MMBPR** module coordinates part-based and global feature refinement. The **VLCL-MG** module structures the feature space with semantic priors. This integrated design systematically tackles the challenges of part-level representation and spatial reasoning in FGVC.
>
>
> 2.  **Inference Efficiency:** While the training pipeline involves multiple components, the inference process is highly efficient. Only the final stage of the global branch is used during inference, allowing us to completely remove the part branch and the CLIP model. The computational cost is primarily from the backbone and a small number of additional layers (one ReSAF and three Encoder Layers). Specifically, without accounting for I/O and using an RN50 backbone with a hidden dimension of 768, our model achieves an inference speed of **18.3 ms/image (54.6 FPS)** with a batch size of 1, and **3.1 ms/image (321.8 FPS)** with a batch size of 8 on a single RTX 4090 GPU, demonstrating strong practicality.
>
> 3.  **Significance of Improvement:** The absolute performance gains, while seemingly modest, are significant given the high baseline and inherent difficulty of distinguishing extremely similar sub-categories in FGVC. Achieving further improvements on these challenging benchmarks is notably difficult. More importantly, our method demonstrates consistent and robust State-of-The-Art (SOTA) performance across **five different datasets** and both **CNN and Transformer backbones**, indicating strong generalization ability. Furthermore, FGVC is known to be sensitive to hyperparameters [1]. Our reported results were achieved with only a single, simple set of hyperparameters to prioritize a generalizable algorithm. As shown below, targeted hyperparameter tuning can lead to even higher performance, suggesting our method's potential is not fully capped by the reported numbers.
>
>     *Example on CUB (ViT-B): SOTA 92.3% → 93.0%*
>     | Hidden Dimension | Accuracy (%) |
>     |------------------|--------------|
>     | 512              | 92.76        |
>     | 768              | 92.34        |
>     | 1024             | 92.30        |
>
>     | $\tilde{\epsilon}_s$       | Accuracy (%) |
>     |--------------------------|--------------|
>     | [0.1, 0.1, 0.2, 1.0]     | 92.99        |
>     | [0.1, 0.1, 0.1, 1.0]     | 92.93        |
>     | [0.1, 0.2, 0.4, 1.0]     | 92.76        |
>
>     *Example on AIR (ViT-B): SOTA 96.5% → 96.7%*
>     | Hidden Dimension | Accuracy (%) |
>     |------------------|--------------|
>     | 512              | 96.69        |
>     | 768              | 96.48        |
>     | 1024             | 96.60        |
>
>     | $\tilde{\epsilon}_s$       | Accuracy (%) |
>     |--------------------------|--------------|
>     | [0.1, 0.1, 0.2, 1.0]     | 96.72        |
>     | [0.1, 0.1, 0.1, 1.0]     | 96.66        |
>     | [0.1, 0.2, 0.4, 1.0]     | 96.69        |
>
> ```
> [1] Rethinking Binary Hyperparameters for Deep Transfer Learning
> ```

---

> > ### Author Response · Authors · 2025-11-24
> >
> > **Q2: Is the number of part texts predetermined? How does the method adapt to tasks with varying part texts across categories?**
> >
> > **A2:** The part texts are predetermined based on the common semantic parts of the coarse-grained category (e.g., "bird" for CUB, "plane" for AIR), as listed in Appendix C. This design effectively guides the model to focus on discriminative regions.
> >
> > 1.  **Handling Multiple Coarse-Grained Categories:** For a classification task containing multiple distinct coarse-grained categories (e.g., a dataset with both birds and planes), the number of part texts would naturally differ across these categories. To handle **this situation where the number of part texts varies**, we recommend a two-stage approach: first, a coarse classifier identifies the main category, and then a dedicated PSCL model, trained specifically for that category (and its corresponding set of part texts), performs fine-grained classification. This aligns with practical scenarios.
> >
> > 2.  **Robustness to Occlusion/Variation:** Our method is robust to another scenario that can lead to **a variation in the *effective* number of part texts** per image, such as when certain parts are occluded or absent. Equation 6 in the paper applies a mask to image features, effectively ignoring gradients from missing parts during training. During inference, only the global branch is used, so the part branch has no impact on the final result.
> >
> > 3.  **Impact of Part Text Quantity:** We conducted experiments to analyze the effect of the number of part texts (`N`). The results below suggest that 4-6 part texts offer a good balance between performance and complexity. Using even a single part text (e.g., "bird") still yields competitive results, as other effective components (like progressive learning) contribute significantly. While using more parts increases computational cost, the performance gains diminish marginally. The experiments were performed using RN50.
> >
> > | Part text| N | Accuracy % |
> > | - | - | - |
> > | background of a plane, tail of a plane, Logo of a plane, engine of a plane, landing gear of a plane, fuselage of a plane | 6 | 95.14  |
> > | background of a plane, tail of a plane, head of a plane, fuselage of a plane  | 4 | 94.74  |
> > | background of a plane, plane | 2 | 94.75 |
> > | plane | 1 | 94.66  |
> >
> >
> > | Part text| N | Accuracy % |
> > | -| - | -- |
> > | background of a bird, head of a bird, foot of a bird, body of a bird, mouth of a bird | 5 | 89.13|
> > | background of a bird, head of a bird, body of a bird| 3 | 88.83|
> > | background of a bird, bird| 2 | 88.47|
> > | bird | 1 | 88.44|

---

> > > ### Author Response · Authors · 2025-11-24
> > >
> > > **Q3: Can large language models (LLMs) accurately describe subtle differences between categories?**
> > >
> > > **A3:** We appreciate the reviewer's question. We would like to clarify a potential misunderstanding: **our method does not use LLMs to describe fine-grained visual differences or to perform image recognition.**
> > >
> > > 1.  **Precise Role of the LLM:** The LLM was used **exclusively to generate intermediate category names** (e.g., generating "Tern" as a semantic midpoint between two specific bird species). This process, detailed in the third paragraph of Section 3.3 and exemplified in Appendix E, helps construct a semantically enriched label space for contrastive learning. It was **not employed** to generate part-level descriptions or to analyze image content. We acknowledge that the description in Appendix A may have been ambiguous; this has been clarified in our revised manuscript.
> > >
> > > 2.  **Exploration of Foundational Models in FGVC:** We conducted experiments to assess the capability of large models for the FGVC task. As shown below, fine-tuning Qwen2.5-VL yields results significantly inferior to those of specialized, smaller models:
> > >     | Dataset | AIR| CUB| CAR| DOG|
> > >     |-|-|-|-|-|
> > >     | Qwen2.5-VL | 0.85129 | 0.91763 | 0.94887 | 0.90306 |
> > >
> > >     We attribute this to two main reasons. First, as also indicated in [2], CLIP's visual encoder itself is suboptimal for FGVC when fine-tuned directly, as it sacrifices fine-grained discriminability for broad semantic alignment. Our own fine-tuning of the CLIP encoder confirms this:
> > >     | Dataset | AIR| CUB| CAR| DOG|
> > >     |-|-|-|-|-|
> > >     | CLIP-ViT | 0.77827 | 0.89575 | 0.93247 | 0.84650 |
> > >
> > >     Second, consistent with findings in [3], a gap exists between the factual knowledge stored in LLMs and their visual recognition capabilities for fine-grained concepts. For instance, while an LLM can textually describe that a "T-tail" is a characteristic of a Boeing 727, it often fails to recognize a Boeing 727 from an image of a T-tail.
> > >
> > > 3.  **PSCL's Approach:** Therefore, in PSCL, we do not directly use the CLIP visual encoder or LLMs for the core visual recognition task. Instead, they serve only as **auxiliary tools**—CLIP provides initial guidance for locating potential regions of interest, and the LLM helps generate a better-structured label space for initial clustering. The primary recognition capability is derived from our proposed contrastive learning framework trained on the target dataset.
> > >
> > > ```
> > > [2] Learning transferable visual models from natural language supervision
> > > [3] ANALYZING AND BOOSTING THE POWER OF FINE GRAINED VISUAL RECOGNITION FOR MULTI-MODAL LARGE LANGUAGE MODELS
> > > ```

---

> > > > ### Comment · Reviewer_WQPm · 2025-11-26
> > > >
> > > > Thanks for the responses, authors have addressed my concerns.

---

> > > > > ### Author Response · Authors · 2025-11-27
> > > > >
> > > > > Thank you very much for your thoughtful comments and careful evaluation of our work. We are glad to hear that your concerns have been fully addressed. We sincerely appreciate your constructive feedback, which has helped us further improve the clarity and quality of the manuscript.

---

### Official Review · Reviewer_E1mk · 2025-10-29

**Soundness:** 3
**Presentation:** 3
**Contribution:** 3
**Rating:** 8
**Confidence:** 4

**Summary:**

The paper proposes PSCL for fine-grained visual classification, decomposing the problem into three parts: (i) a PLM that uses ClearCLIP text prompts to localize part regions and decouple selection from encoding; (ii) MMBPR for multi-scale, multi-branch progressive reasoning with ReSAF (a reverse-key scale-attention fusion) to reduce cross-scale redundancy; (iii) VLCL-MG that aligns image features with coarse/mid/fine textual concepts. Results are reported on AIR/CAR/CUB/NABirds/DOG with RN50/ViT-B/Swin-B. The masking pipeline is “argmax → morphological refinement,” and a component-level GFLOPs table is provided. The abstract claims an anonymous code link.

**Strengths:**

Clear problem decomposition. The “where to look / how to represent / how to align” split matches FGVC pain points and keeps the design modular.

Interpretability. The PLM produces intuitive part masks from ClearCLIP similarity maps; visualizations help readers see what the model attends to.

Solid component study. ReSAF has reasonable motivation and ablations against MLP / cross-attention baselines; gains are consistent rather than cherry-picked.

Breadth of evidence. Improvements hold across multiple datasets and backbones, suggesting the method isn’t overly tuned to a single regime.

This separation also makes the ablations easier to interpret.

**Weaknesses:**

“Reverse” label smoothing in Eq. (17). The formula appears to assign a smaller weight to the true class and larger to others, which is the opposite of standard label smoothing. I suspect this might be a typo; if it’s intentional, please explain the intuition and provide a small ablation.

No end-to-end compute/latency. The paper lists component GFLOPs (and notes branch-count growth), but omits whole-model FLOPs/parameter count/peak VRAM / latency (e.g., batch=1 & batch=8). That makes deployment comparisons hard. If I missed this in the appendix, please point me to the exact page/table.

Reproducibility inconsistency. The abstract links to an anonymous repo, yet the Reproducibility Statement says “code available upon request.” Please unify this to an anonymous public repo with full training/eval scripts, configs, and (ideally) weights. If this is already in the appendix, please point me to the exact page/table.

PLM details are insufficient for reproduction. To replicate the masks: specify ClearCLIP version/checkpoint and input resolution; provide the full part-prompt lists per dataset; describe kernel shape/size/iterations for morphology; state whether similarity logits are normalized or temperature-scaled before argmax.

Mid-level text generation is under-specified. For datasets without native hierarchies, the mid-level labels are LLM-generated. Please release prompts/sampling & screening rules, and include a broader noise-sensitivity study beyond one dataset.

Tuning fairness. Most hyperparameters (except LR) are tuned on AIR+RN50 and transferred elsewhere. Add an “equal-budget tuning” table or a light per-dataset retune to rule out hidden advantages.

Minor writing issues. There are a few typos (e.g., “TWe utilize…”). Figure 2 is dense; consider trimming labels and adding a one-page “Training Recipe.”

**Questions:**

Is the reverse smoothing in Eq. (17) intentional? If yes, provide intuition (e.g., effect on hard examples/confidence distribution) and a small ablation on AIR & CUB comparing standard vs. reverse smoothing, including interaction with γ.

Provide an end-to-end resource table for RN50 / ViT-B / Swin-B × number of branches N: FLOPs, params, peak VRAM, and latency at B=1 and B=8. Include comparisons to TransFG/CSQA-Net and a strong CLIP-head baseline (e.g., CoOp/CoCoOp/Tip-Adapter) under matched resolution/budget.

PLM reproducibility: exact ClearCLIP release/ckpt, input resolution; complete part-prompt lists; morphology kernel/iterations; whether logits are normalized or temperature-scaled pre-argmax; plus a short sensitivity study.

Mid-level text protocol & robustness: release prompts, sampling/filters, and the final mid-level lists for AIR/CAR/CUB/DOG; quantify performance variance across multiple LLM samples to bound noise.

ReSAF intuition: add a histogram or case study of cross-scale similarities to show how the “reverse key” steers attention away from redundant regions rather than only relying on Table-level gains.

Hyperparameter fairness: provide an equal-budget tuning comparison or justify why transferring AIR-tuned hypers doesn’t bias outcomes.

---

> ### Author Response · Authors · 2025-11-24
>
> We sincerely thank the reviewer for the thorough and constructive assessment. We address all questions and concerns below, and we have incorporated or are incorporating the corresponding revisions into the updated manuscript and appendix.
>
> ---
>
> **Q1: Clarification of Eq. (17) (“reverse” label smoothing).**
>
> **A1:** We appreciate the reviewer for catching this issue. The equation in the original submission contains a typo. In our notation, $\epsilon_s$ represents the stage-dependent smoothing confidence, which increases the confidence of the true label as $s$ progresses; however, the notation unintentionally conflicted with the standard label-smoothing convention in which the *smoothing noise factor* is denoted by $\epsilon_s$. To avoid confusion, we corrected the notation throughout the revised manuscript.
>
> The corrected formulation is:
>
> $
> \tilde{y} _ {s,n',c}=
> \begin{cases}
> 1 - \epsilon_s, & \text{if } c = y _ {s,n'}, \\\\
> \epsilon_s / (C-1), & \text{otherwise},
> \end{cases}
> \tag{17}
> $
>
> where $\epsilon_s$ gradually decreases as the stage index $s$ increases, enabling the MMBPR module to generate progressively more confident predictions.
>
> ---
>
> **Q2: End-to-end compute, VRAM, and latency.**
>
> **A2:** We agree that the original version lacked complete end-to-end resource metrics. We have now added full training and inference statistics using a single NVIDIA RTX-4090 GPU for all backbones and datasets (hidden dimension = 768; 4 stages; batchsize = 16). The full table is included in the appendix; below are the main results:
>
> **Training time and peak VRAM**
>
> | Model  | Dataset | Train Time (h) | Peak VRAM (GB) |
> | ------ | ------- | -------------- | -------------- |
> | RN50   | AIR     | 3.7            | 18             |
> | RN50   | CAR     | 3.8            | 15.1           |
> | RN50   | CUB     | 3.0            | 16.5           |
> | RN50   | NAB     | 12             | 16.5           |
> | RN50   | DOG     | 5.5            | 15.1           |
> | ViT-B  | AIR     | 5.1            | 22             |
> | ViT-B  | CAR     | 5.4            | 19.1           |
> | ViT-B  | CUB     | 4.2            | 20.6           |
> | ViT-B  | NAB     | 16.6           | 20.6           |
> | ViT-B  | DOG     | 8.0            | 19.1           |
> | Swin-B | AIR     | 4.8            | 23.8           |
> | Swin-B | CAR     | 5.1            | 21.6           |
> | Swin-B | CUB     | 3.9            | 22.7           |
> | Swin-B | NAB     | 15.4           | 22.7           |
> | Swin-B | DOG     | 7.5            | 21.6           |
>
> **Inference-time policy.**
> PSCL follows the standard practice of multi-branch FGVC models: only the *global branch* from the final stage is used at test time. Thus, all part-level branches and ClearCLIP are removed. For 4 stages, inference computation reduces to:
> **backbone + one ReSAF + three encoder layers.**
>
> The actual inference comparison (per-image time in ms) is as follows:
>
> | Batch | Model  | **PSCL** | Backbone |
> | ----- | ------ | -------- | -------- |
> | 1     | ViT-B  | 17.26    | 6.39     |
> | 1     | RN50   | 18.29    | 6.56     |
> | 1     | Swin-B | 31.41    | 21.75    |
> | 8     | ViT-B  | 5.18     | 3.97     |
> | 8     | RN50   | 3.11     | 1.38     |
> | 8     | Swin-B | 5.01     | 3.45     |
>
> Throughput (images/s):
>
> | Batch | Model  | **PSCL** | Backbone |
> | ----- | ------ | -------- | -------- |
> | 1     | ViT-B  | 57.94    | 156.59   |
> | 1     | RN50   | 54.66    | 152.47   |
> | 1     | Swin-B | 31.84    | 45.98    |
> | 8     | ViT-B  | 193.13   | 251.91   |
> | 8     | RN50   | 321.89   | 724.25   |
> | 8     | Swin-B | 199.79   | 289.70   |
>
> The table below reports the peak GPU memory usage during inference for different model architectures and batch sizes, measured in GB.
>
> | Model  | Batch Size 1 | Batch Size 8 |
> | ------ | ------------ | ------------ |
> | PSCL(RN50)   | 3.65         | 3.93         |
> | PSCL(ViT-B)  | 4.54         | 4.78         |
> | PSCL(Swin-B) | 4.71         | 5.04         |
>
>
> We have added these results to the revised manuscript, in Section 4.4 TRAINING AND INFERENCE EFFICIENCY and Appendix B. Additionally, we have included a description of the inference process:
> "
> At inference time, the prediction from the final stage is utilized, and the inference strategy relies solely on the global branch:
>
> $
>     \mathbf{P}^{\text{global}}=\mathbf{P} _ {s=4,\, n'=0},
> $
>
> in which ClearCLIP and redundant part-level branches are removed during inference, thereby enabling substantially faster computation.
> "

---

> > ### Author Response · Authors · 2025-11-24
> >
> > **Q3:  Reproducibility and anonymous code release.**
> >
> > **A3:** We confirm that the anonymous repository referenced in the abstract contains full training/evaluation scripts, configuration files, and dataset processing rules. Pretrained weights cannot be uploaded anonymously under current constraints, but all results can be reproduced with the provided code.We believe that the misunderstanding may have arisen from the wording in the Reproducibility Statement. In the revised version, we changed it to:
> > "We release the code, trained model checkpoints, and the dataset as anonymous supplementary materials, enabling other researchers to reproduce our experiments under the same settings."
> >
> > 1. **ClearCLIP details.**
> > ClearCLIP currently has only one stable release; it is derived by removing CLIP’s final components without retraining, and therefore shares weights with CLIP. In all experiments, we use CLIP ViT-B, which is also the recommended configuration of ClearCLIP.
> >
> > 2. **Input resolution.**
> > The Implementation Details section lists all resolutions. For ViT-B, careful tuning showed that 518×518 with ViT-B/14 yields the strongest FGVC performance; we explain the resolution selection and fairness considerations in the revised Appendix H.
> >
> > 3. **Part-prompt lists.**
> > Complete lists per dataset are now included in Appendix C.
> >
> > 4. **Morphological operations.**
> > We use a 3×3 kernel, with one dilation followed by one erosion (closing). This has been clarified in the revised version, with an update to Equation 5.
> >
> > 5. **On the use of temperature scaling.**
> > Similarity tensor $\mathbf{S} $ is produced by Eq. (2); Eq. (3) simply takes the argmax across the part dimension to form a mask. Since normalization or temperature scaling does not affect the outcome of Eq. (3), we do not apply them. We have tried substituting the mask with temperature-scaled $\mathbf{S} $ to directly weight features, but this consistently decreases accuracy (−1.3% on AIR with RN50), and this result is consistent with our motivation that:
> > ***Shared-region matching emphasizes cross-category consistency, which may conflict with capturing subtle within-part discriminative details essential for fine-grained recognition.***
> >
> > ---
> >
> > **Q4: Mid-level concept generation and robustness**
> >
> > **A4:** We have already added example mid-level concepts in Appendix E. The complete list (1171 total categories) cannot be included in the paper due to space limitations, but it is available in the dataset implementation for reproducibility.
> >
> > **LLM protocol.**
> > We use ChatGPT-4o with the following prompt (AIR example):
> >
> > > “Please classify the following fine-grained categories based on visually discernible characteristics. Each category must belong to two distinct intermediate-grained categories. Output one line per category:
> > > [Fine-grained], [Intermediate-1], [Intermediate-2]:
> > > 707-320 727-200 737-200 …”
> >
> > Based on your suggestion, the LLM protocol is now documented in the Appendix F.
> >
> > **Noise-sensitivity experiments.**
> > We additionally evaluated **DeepSeek-R1** and **Qwen-2.5-Max** under greedy decoding:
> >
> > | Setting                           | Model        | Accuracy |
> > | --------------------------------- | ------------ | -------- |
> > | Baseline                          | ChatGPT-4o   | 95.14    |
> > | Re-label using same intermediate-grained categories | DeepSeek-R1  | 95.08    |
> > |                                   | Qwen-2.5-Max | 95.11    |
> > | Re-generate intermediate-grained categories    | DeepSeek-R1  | 94.83    |
> > |                                   | Qwen-2.5-Max | 95.21    |
> >
> > Only DeepSeek-R1’s regenerated intermediate-grained categories became overly coarse (engine type, wing type), but LLM variation overall shows robustness, and all models outperform the version without VLCL-MG (94.54%).
> >
> > Based on your suggestion, we added noise-sensitivity experiments and discussion to the Appendix G.

---

> > > ### Author Response · Authors · 2025-11-24
> > >
> > > **Q5: Hyperparameter fairness and transferability**
> > >
> > > **A5:** We acknowledge that FGVC hyperparameters are sensitive (as noted in prior work [1]). Our goal is to demonstrate strong *generalization* rather than per-dataset tuning. While AIR+RN50 is the only configuration for which we performed grid search, we provide additional evidence showing that light retuning can further improve performance:
> > >
> > > **CUB with ViT-B**
> > >
> > > | Hidden Dim | Accuracy  |
> > > | ---------- | --------- |
> > > | 512        | **92.76** |
> > > | 768        | 92.34     |
> > > | 1024       | 92.30     |
> > >
> > > | $ \tilde{\epsilon}_s $ | Accuracy  |
> > > | ---------------------- | --------- |
> > > | [0.1, 0.1, 0.2, 1.0]   | **92.99** |
> > > | [0.1, 0.1, 0.1, 1.0]   | 92.93     |
> > > | [0.1, 0.2, 0.4, 1.0]   | 92.76     |
> > >
> > > **AIR with ViT-B**
> > >
> > > | Hidden Dim | Accuracy  |
> > > | ---------- | --------- |
> > > | 512        | **96.69** |
> > > | 768        | 96.48     |
> > > | 1024       | 96.60     |
> > >
> > > | $\tilde{\epsilon}_s $ | Accuracy  |
> > > | ---------------------- | --------- |
> > > | [0.1, 0.1, 0.2, 1.0]   | **96.72** |
> > > | [0.1, 0.1, 0.1, 1.0]   | 96.66     |
> > > | [0.1, 0.2, 0.4, 1.0]   | 96.69     |
> > >
> > > We added guidance summarizing these observations:
> > >
> > > 1. For deeper or more globally receptive backbones, reduce early-stage smoothing and use a smaller hidden dimension.
> > > 2. For datasets with many hard examples, slightly increase $ \gamma $ (≤6).
> > >
> > > ```
> > > [1] Rethinking Binary Hyperparameters for Deep Transfer Learning
> > > ```
> > >
> > > ---
> > >
> > > **Q6: ReSAF intuition and visual analysis**
> > >
> > > **A6:** ReSAF is designed such that deep features dominate classification, while shallow features complement them. The reversed key directs attention to regions under-attended by deep features. Cross-scale feature interaction is handled later in MMBPR’s Eq. (11) via concatenation followed by learnable selection in encoder layers.
> > >
> > > We will add qualitative case studies in the Appendix I to illustrate the effect of the reversed-key mechanism. Visualizations of the attention mechanisms for several ReSAF cases have been included in Figure 5 of the revised version.
> > >
> > >
> > >
> > > ---
> > >
> > > **Q7: Writing, typos, and figure clarity**
> > >
> > > **A7:** All typo (including “TWe utilize…”) have been corrected.
> > > We have updated the textual description to facilitate a better understanding of Figure 2, and the simplification of the figure is currently in progress.

---

> ### Comment · Reviewer_E1mk · 2025-11-26
>
> The authors have addressed my concerns.

---

> > ### Author Response · Authors · 2025-11-27
> >
> > Thank you very much for your positive assessment.
> > We appreciate your careful review and constructive feedback throughout the process, which has helped us improve the clarity and quality of the manuscript.

---

### Author Response · Authors · 2025-12-04
**Summary of Comments**

Dear ACs,

We would like to begin by expressing our sincere appreciation for the ACs’ time, attention, and thoughtful efforts during the review process.

During the rebuttal stage, we provided comprehensive, point-by-point responses to all reviewer comments and substantially improved the manuscript. Importantly, **before the deanonymization incident**, two reviewers had already updated their overall assessments based on our rebuttal and revisions—**Reviewer WQPm (4 → 6)** and **Reviewer UrRj (6 → 8)**—and **Reviewer E1mk indicated that they had no remaining concerns**. The overall scores before the incident are **8864**.

These updates were made **entirely prior** to the leak and its subsequent rollback. The updates that motivated the higher scores are already fully reflected in the current PDF and rebuttal: we corrected formulas and figures that could lead to ambiguity; provided a complete and rigorous description of both training and inference costs; clarified and expanded technical details including the inference pipeline, text generation procedure, the exact realization of morphological closing, and the role of the LLM; offered in-depth explanations of the design motivations and functional roles of ReSAF, morphological operations, and MMBPR; added extensive noise-robustness experiments, hyperparameter studies, and ablation analyses; and incorporated more visualizations.

We also wish to clarify that **the concerns raised by Reviewer fUYe stemmed from misunderstandings of our model and from differences in task definition**, and we have addressed them carefully in both the rebuttal and the revised manuscript. In brief, we clarified the expected nature of part-based gains, the task-mismatch explanation regarding iNaturalist, and the adequacy of our reported resolution and efficiency settings.

We fully respect the decision to revert the reviews for fairness. This note is provided solely to help the ACs interpret the current reviews in the context of the clarified and strengthened manuscript.

Best Regards,
Authors

---

Below, we provide a structured summary of the main clarifications and improvements made during the rebuttal stage.

---

### **Summary of Reviewer Comments and Our Responses**

**1. Clarification of methodological details and potential misunderstandings**

We substantially refined the descriptions of PLM, MMBPR, and ReSAF, clarifying multi-level feature extraction in ViT, the precise operation of the reversed-key cross-scale attention, and the role of morphological closing in Eq. (5). We also resolved misunderstandings regarding the use of LLMs: they are **not** used for visual recognition or describing fine-grained visual cues, but only to generate intermediate category names. Ambiguous equations, figures, and terminology have been corrected.

**2. Reproducibility, computational transparency, and inference design**

The revised version includes complete implementation details for mask generation, mid-level text construction, hyperparameters, ClearCLIP settings, and morphology kernels. We added full end-to-end training and inference statistics (FLOPs, VRAM, latency, throughput) and clarified that **inference relies solely on the final global branch**, making the method efficient despite multi-branch training.

**3. Assessment of component effectiveness and performance gains**

We provided expanded ablations on part-number variation, ReSAF variants, hyperparameters, and additional noise-sensitivity experiments. In particular, we conducted systematic studies on (1) how different numbers and configurations of part texts affect performance; (2) multiple ReSAF variants to highlight the contribution of cross-level attention and the reversed-key mechanism; and (3) the stability of key hyperparameters across datasets through light retuning experiments. We further added additional visualizations—including ReSAF attention maps, part-score distributions to illustrate how our components reduce feature redundancy, enhance spatial awareness, and provide more discriminative representations, especially for hard examples.

**4. Task definition, applicability, and dataset differences**

We clarified that limitations on heterogeneous datasets (e.g., iNaturalist) arise from **task-definition mismatch** rather than scalability issues, as such datasets do not share consistent part vocabularies or hierarchical semantics. Even under restricted settings, our model performs competitively, and adding richer semantic structure remains a key direction for future extension.

**5. Resolution fairness**

All input resolutions were clearly listed, and we added resolution-sensitivity experiments to ensure fairness. As modern FGVC pipelines use cropping and ROI-enlargement, resolution alone is not a sufficient fairness metric. We therefore also include full computational cost analyses and detailed resolution settings and results to give a more accurate view of practical performance.

---

### Public Comment · ~Zhijian_Lin2 · 2026-03-01

Dear Area Chair and Reviewers,

We sincerely thank the Area Chair and all reviewers for the insightful comments and constructive discussions. Your feedback has been invaluable in improving the clarity and completeness of our work, and has greatly contributed to the final camera-ready version of the paper.

In the revised manuscript, we have carefully incorporated the AC’s suggestions. In Section 3.2, we explicitly clarify the similarities and differences between our PSCL framework and PART [1], and further elaborate on the motivation behind our design choices. This revision better positions our method with respect to prior work and highlights its conceptual distinctions.

We have also expanded the discussion on input resolution. In the *Comparison with Other Methods* section, we now report the input resolutions of different models to facilitate direct comparison and ensure fairness and transparency in evaluation.

In addition, we have added a *Discussions* section to briefly clarify the main limitations of PSCL. Specifically, we note that the framework is constrained by task definition and is primarily suited for single-domain fine-grained classification where semantic parts can be consistently defined. Moreover, due to the coarse granularity of the current part localization mechanism, performance on datasets with subtle and fine-grained components can be affected by input resolution.

We sincerely appreciate the reviewers’ careful reading and valuable suggestions, which have significantly strengthened the final version of our paper.

Best regards,

Authors

```
[1] Zhao et al., "Part-guided Relational Transformers for Fine-grained Visual Recognition," IEEE TIP 2021.
```

---

### Meta-Review · Area_Chair_2mig · 2025-12-23

**Summary:**

This paper proposes part-level semantic contrastive learning for fine-grained image classification. This is a very classic and traditional field, and part localization is widely used in fine-grained recognition methods. Due to the special circumstances of ICLR, the initial score of this paper was slightly lower than the borderline acceptance. Based on the comments from the reviewers, the authors made extensive revisions and improvements in the method sections of sections 3.2 and 3.3.

**Reviewer Concerns:**

+ Among them, reviewer WQPm raised questions about the method design, improvements, and details. AC carefully read the original and revised versions, and these shortcomings were indeed improved to some extent.

+ Reviewer fUYe raised the similarities and differences between the paper [1] Part-guided Relational Transformers for Fine-grained Visual Recognition. AC is familiar with these related works and does think there are certain similarities, including the use of part localization and transformer embedding. AC also noted the performance limitations of this paper, including the use of classic fine-grained datasets, and the evaluation of more and broader fine-grained datasets, or the generalization ability under small sample conditions, which still need to be explored. The AC requested the authors to add more discussion and differentiation from the aforementioned work [1] in the final version. Although this paper incorporates multi-granularity contrastive learning methods such as CLIP, it still requires more detailed explanation comparing to conventional works only using ResNet or ViT backbones..

+ Regarding the final response, the authors and reviewer fUYe suggested that the authors update the basic resolution information in the main text to ensure fairness, based on tests on larger datasets, including iNatureList and resolution reports. iNatureList does indeed show that the method has clear limitations, and the authors need to explain and analyze these limitations in the limitations section, or add more discussion.

In summary, this paper is generally at a very borderline level. During the rebuttal process, it did receive some positive feedback and resolved some issues regarding technical details. However, the AC also acknowledged reviewer fUYe's comments, believing that the authors should make corrections. Considering the paper's rating and the fact that these shortcomings can be corrected in the final version.

**Reviewer Scores:**

Reviewer WQPm would raise the score from 4 to 6, while fUYe remains a negative score of 4. The final score would be 3 positive score and 1 negative one.

During the discussion, three reviewers leaned towards a positive view, while one reviewer leaned towards a negative view, which put the paper on the verge of acceptance.

---

### Decision · Program_Chairs · 2026-01-26

Accept (Poster)